# Uncertainty-Calibrated Prediction of Randomly-Timed Biomarker Trajectories with Conformal Bands

Vasiliki Tassopoulou[1,*]    Charis Stamouli[2,*]
Haochang Shou[3]    George J. Pappas[2]    Christos Davatzikos[1]

[1] Center for AI and Data Science for Integrated Diagnostics (AI2D), Perelman School of Medicine, University of Pennsylvania, Philadelphia, PA, USA
[2] Department of Electrical and Systems Engineering, University of Pennsylvania, PA, USA
[3] Department of Biostatistics and Epidemiology, University of Pennsylvania, PA, USA
E-mail: vtass@seas.upenn.edu, stamouli@seas.upenn.edu, hshou@pennmedicine.upenn.edu, pappasg@seas.upenn.edu, christos.davatzikos@pennmedicine.upenn.edu

## Abstract

We introduce a novel conformal prediction framework for constructing conformal prediction bands with high probability around biomarker trajectories observed at subject-specific, randomly-timed follow-up visits. Existing conformal methods typically assume fixed time grids, limiting their applicability in longitudinal clinical studies. Our approach addresses this limitation by defining a time-varying nonconformity score that normalizes prediction errors using model-derived uncertainty estimates, enabling conformal inference at arbitrary time points. We evaluate our method on two well-established brain biomarkers—hippocampal and ventricular volume—using a range of standard and state-of-the-art predictors. Across models, our conformalized predictors consistently achieve nominal coverage with tighter prediction intervals compared to baseline uncertainty estimates. To further account for population heterogeneity, we develop group-conditional conformal bands with formal coverage guarantees across clinically relevant and high-risk subgroups. Finally, we demonstrate the clinical utility of our approach in identifying subjects at risk of progression to Alzheimer's disease. We introduce an uncertainty-aware progression metric based on the lower conformal bound and show that it enables the identification of 17.5% more high-risk subjects compared to standard slope-based methods, highlighting the value of uncertainty calibration in real-world clinical decision making. We make the code available at github.com/vatass/ConformalBiomarkerTrajectories.

## 1 Introduction

Predicting biomarker trajectories is crucial for monitoring disease progression and improving patient outcomes through early intervention. In neurodegenerative diseases such as Alzheimer's, volumetric biomarkers—most notably hippocampal atrophy—are closely associated with clinical decline and are widely used to track disease progression [1–3]. Given the clinical importance of biomarker trajectories, recent advances in machine learning have led to a growing number of methods designed to forecast biomarker evolution from longitudinal data [4–8]. However, inter-individual variability in disease progression and measurement noise inherent in imaging data make predictions inherently uncertain—models cannot determine exact future values with complete confidence. If uncertainty

---

*Equal contribution

39th Conference on Neural Information Processing Systems (NeurIPS 2025).

is not properly accounted for, predictions can cause high-risk subjects to be misclassified as stable. Such potential misclassifications hinder the deployment of biomarker predictors in clinical workflows, where failure to identify high-risk individuals can delay urgently needed intervention or exclude such individuals from trials. To enable safe and reliable decision-making in healthcare, uncertainty calibration of the learned predictors is important.

A popular approach for equipping predictors with reliable uncertainty estimates is given by conformal prediction [9–11]. Conformal prediction is highly versatile, accommodating any black-box predictive model (e.g., neural networks) and any data distribution (e.g., non-Gaussian). The first step in the approach is to evaluate the learned model's prediction error, as characterized by a so-called *nonconformity score*, on a held-out calibration dataset. Assuming that the calibration and test data are sampled independently from the same probability distribution*, the prediction error on the calibration data can be used to estimate the prediction error on the test data. Thus, model predictions can be turned into regions that are guaranteed to cover the true observations with arbitrarily high probability.

In the context of trajectory prediction, existing conformal methods have focused on fixed-time trajectories [12–16]. However, biomarker datasets typically contain randomly-timed biomarker trajectories due to missed visits and variable scheduling in clinical studies (see Figure 1). Such trajectories create a more complex setting for designing conformal prediction bands with guaranteed coverage properties, which are critical for safely delivering early diagnoses and effectively guiding clinical interventions.

In this paper, we propose a novel conformal prediction method tailored to the setting of randomly-timed biomarker trajectories. Our approach derives prediction bands that are guaranteed to cover the entire true biomarker trajectory with arbitrarily high probability. We apply our method to forecasting hippocampal and ventricular-volume trajectories—two well-established biomarkers to monitor the onset and progression of Alzheimer's disease (AD). To better capture variability across clinically meaningful subgroups, including high-risk individuals such as those diagnosed with Mild Cognitive Impairment (MCI), we design group-conditional conformal predictors that offer formal coverage guarantees within covariate-defined strata. We

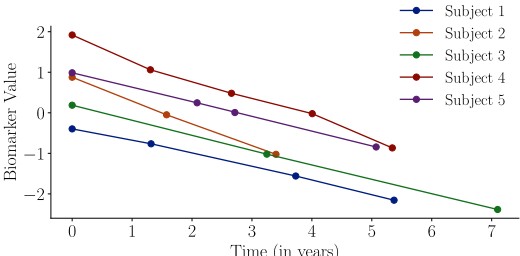

Figure 1: Illustration of randomly-timed biomarker trajectories from five subjects. Time is measured with respect to the first biomarker observation of each subject. Note the varying trajectory length and the different time points of observations across subjects.

evaluate these predictors across five key stratifications—sex, race, diagnosis, education level, and APOE4 allele status—motivated by clinical risk and the need for equitable performance across underrepresented demographic subgroups. Beyond evaluating calibration, we translate conformal uncertainty into a downstream decision task: identifying MCI patients at high risk of progression to Alzheimer's disease. We introduce a trajectory-based risk score derived from the lower conformal bounds and show that this uncertainty-calibrated metric enables more inclusive and safety-aware subject selection in scenarios such as trial enrichment or early intervention planning.

Our contributions can be summarized as follows: **1)** We introduce a new conformal prediction framework for randomly-timed biomarker trajectory forecasting. Our method defines a nonconformity score over multiple time points—similar in spirit to Yu et al. [17], Cleaveland et al. [18]—with the key distinction that our time points are not fixed across individuals but randomly distributed, reflecting real-world clinical follow-up schedules. **2)** We test our approach on two well-established neurodegenerative biomarkers—hippocampal volume and ventricular volume—using real clinical data. By conformalizing a range of standard and state-of-the-art predictors, we demonstrate that our method achieves the nominal coverage while maintaining tight prediction bands, outperforming baselines that rely solely on model-specific uncertainty estimates. **3)** To address population heterogeneity, we incorporate covariate-based stratification into the calibration step and develop group-conditional conformal predictors with formal subgroup-level coverage guarantees. We empirically validate these guarantees across five demographic and clinically relevant stratifications: sex, race, diagnosis, education level, and APOE4 allele status. **4)** We demonstrate the clinical utility of our

---

*Conformal methods typically require that the calibration and test data are *exchangeable*—a milder condition than that of independent and identically distributed data (see Appendix A).

uncertainty-calibrated conformal prediction bands in a downstream decision task, where the goal is to identify MCI subjects at high risk of progression to Alzheimer's disease. To this end, we introduce a novel uncertainty-aware progression metric—the rate of change bound—which leverages conformal uncertainty to provide an estimate of the biomarker's rate of change over time. This metric enables the **identification of 17.5% more high-risk MCI subjects** who progress to Alzheimer's disease compared to standard approaches.

## 2 Related Work

**Uncertainty-Calibrated Prediction of Biomarker Trajectories.** Uncertainty-calibrated predictors for biomarker trajectories integrate uncertainty quantification into their architecture. Bayesian methods achieve this by employing probabilistic frameworks to model uncertainty. For instance, Gaussian Processes (GPs) [19] utilize kernel-defined priors over functions to encode structural assumptions, such as smoothness. Their posterior distributions quantify prediction uncertainty through covariance-based inference under Gaussian likelihoods. Several studies [8, 20–24] involve predictors with inherent uncertainty measures based on Gaussian distribution assumptions. In contrast, Lin et al. [25] introduce a conformal variant for fixed-time biomarker trajectories, ensuring asymptotic longitudinal coverage without any distributional assumptions.

**Conformal Prediction for Fixed-Time Trajectories.** Conformal methods for predicting fixed-time trajectories have been developed for settings with single time series data [26–28] and multiple time series data [13, 17, 18, 25, 29, 30]. Our setting differs from the above, involving data with *generally non-temporal* inputs (e.g., sex) and *randomly-timed* trajectory outputs (e.g., hippocampal-volume measurements taken at random time points for each subject). We focus on the latter line of works, whose setting is a special case of ours.

Lin et al. [25] present a conformal prediction method with asymptotic longitudinal coverage guarantees. Stankeviciute et al. [13] and Lindemann et al. [30] design conformal prediction bands with finite-sample coverage guarantees, by leveraging a union bound argument. Sun and Yu [29] propose using copulas to model the uncertainty of predictions at multiple time points. Their approach ensures coverage with narrower prediction bands, but is limited to situations with ample calibration data, as noted by the authors. Yu et al. [17] and Cleaveland et al. [18] overcome this limitation by defining a normalized nonconformity score jointly over multiple time points. Their framework provides narrow prediction bands with simultaneous coverage guarantees over the entire trajectory length. In the next section, we extend their approach to our setting of randomly-timed trajectories, where observations are collected at varying time points.

## 3 Conformal Prediction for Randomly-Timed Trajectories

In this section, we introduce a novel conformal prediction method tailored to settings with randomly-timed trajectories. In Subsection 3.1, we present the problem of conformal prediction for data with vector-valued inputs and randomly-timed trajectory outputs. In Subsection 3.2, we derive conformal prediction bands with coverage guarantees for our setting, inspired by the approach of Yu et al. [17] and Cleaveland et al. [18] for fixed-time trajectories.

### 3.1 Problem Formulation

Our setting of randomly-timed trajectories is characterized by a triplet of random variables $(X, \mathcal{T}, Y)$, where $X \in \mathbb{R}^d$ denotes the input, $\mathcal{T} \subseteq \mathbb{N}_+$ the set of time points at which observations $Y_t \in \mathbb{R}$ are collected, and $Y := \{Y_t : t \in \mathcal{T}\}$ the corresponding trajectory output. In the context of biomarkers, the triplet $(X, \mathcal{T}, Y)$ can be interpreted, for instance, in the following way: i) the input $X$ may involve demographic covariates (e.g., sex, race) as well as the biomarker observation on the patient's first clinical visit, ii) the set $\mathcal{T}$ may include the times of the patient's subsequent visits, and iii) the output $Y$ the corresponding biomarker observations at these times. We note the implicit dependence of the trajectory output $Y$ on the set of time points $\mathcal{T}$.

Suppose we have a dataset $D := \{(X^{(i)}, \mathcal{T}^{(i)}, Y^{(i)})\}_{i=1}^N$ such that for each test example $(X, \mathcal{T}, Y)$, the random variables $(X^{(1)}, \mathcal{T}^{(1)}, Y^{(1)}), \ldots, (X^{(N)}, \mathcal{T}^{(N)}, Y^{(N)})$, and $(X, \mathcal{T}, Y)$ are *exchangeable*[†]. We note that exchangeability is weaker than the standard assumption of independent and identically distributed data, imposing a reasonable condition for biomarker datasets, where each triplet $(X^{(i)}, \mathcal{T}^{(i)}, Y^{(i)})$ corresponds to a different subject. Our goal is to leverage the dataset $D$ and

---

[†]Exchangeability implies that the joint probability distribution of a sequence of random variables remains unchanged under any permutation of the variables (see definition in Appendix A).

the input $X$ to design a high-confidence prediction band for the unknown trajectory $Y$ (see Figure 3). Our problem is formalized in the following statement.

*Problem* 3.1. Consider a given dataset $D$, consisting of $N$ data $(X^{(1)}, \mathcal{T}^{(1)}, Y^{(1)}), \ldots, (X^{(N)}, \mathcal{T}^{(N)}, Y^{(N)})$, and a test example $(X, \mathcal{T}, Y)$, such that all $(X^{(i)}, \mathcal{T}^{(i)}, Y^{(i)})$ and $(X, \mathcal{T}, Y)$ are exchangeable. Given a failure probability $\alpha \in (0, 1)$, design conformal intervals $\mathcal{C}_t(X) \subseteq \mathbb{R}$ for the unknown observations $Y_t$, such that:

$$\mathbf{P}\left(\forall t \in \mathcal{T} : Y_t \in \mathcal{C}_t(X)\right) \geq 1 - \alpha. \tag{1}$$

The intervals $\mathcal{C}_t(X)$ may also depend on the dataset $D$ as well as the parameters $N$ and $\alpha$.

### 3.2 Conformal Prediction Bands for Randomly-Timed Trajectories

In this subsection, we introduce conformal prediction for randomly-timed trajectories, under the setting of Subsection 3.1. Inspired by the conformal methods in [17, 18] for fixed-time trajectories, we design a prediction band that guarantees covering the unknown trajectory $Y$ with arbitrary confidence (see (1)).

We focus on the standard tractable variant of conformal prediction, referred to as split conformal prediction [11]. Specifically, we start by splitting the dataset $D$ into a training set $D_{\text{train}}$ and a calibration set $D_{\text{cal}}$. Let $\mathcal{I}_{\text{train}} := \{i : (X^{(i)}, \mathcal{T}^{(i)}, Y^{(i)}) \in D_{\text{train}}\}$ and $\mathcal{I}_{\text{cal}} := \{i : (X^{(i)}, \mathcal{T}^{(i)}, Y^{(i)}) \in D_{\text{cal}}\}$ denote the corresponding sets of indices. Employing the dataset $D_{\text{train}}$, we train an arbitrary model that maps the input $X$ to predictions $\widehat{Y}_t$ of the future observations $Y_t$ (see Appendix E for examples of predictive models). Moreover, let $\widehat{Y}_t^{(i)}$ denote the estimate of $Y_t^{(i)}$ returned by the learned predictor. The idea is to leverage the data in $D_{\text{cal}}$ and the corresponding predictions $\widehat{Y}_t^{(i)}$ to transform the estimates $\widehat{Y}_t$ into intervals that ensure the guarantee (1). To this end, we define the normalized *nonconformity scores*:

$$R^{(i)} = \max_{t \in \mathcal{T}^{(i)}} \left\{ \frac{|Y_t^{(i)} - \widehat{Y}_t^{(i)}|}{\sigma(\widehat{Y}_t^{(i)})} \right\}, \tag{2}$$

where $\sigma(\cdot) : \mathbb{R} \to \mathbb{R}_+$ is an arbitrary normalizing function, for all $i \in \mathcal{I}_{\text{cal}}$. A natural way of defining the function $\sigma(\cdot)$ is by leveraging some notion of predictive standard deviation (std). In cases of predictors with inherent predictive stds, the function $\sigma(\cdot)$ can be defined accordingly (see Appendix E for examples). Otherwise, the factors $\sigma(\widehat{Y}_t^{(i)})$ can be computed from the given data, employing ideas from [17, 18]. In the following theorem, we use the scores in (2) to design a prediction band composed of intervals $\mathcal{C}_t(X) \subseteq \mathbb{R}$ that ensure the guarantee (1). The proof of the theorem is given in Appendix B.

**Theorem 3.2** (Conformal Prediction Bands for Randomly-Timed Trajectories)**.** *Fix a failure probability $\alpha \in (0, 1)$. Let $\widehat{Y}_t$ be the prediction of the future observation $Y_t$ at time point $t$. Consider the nonconformity scores $R^{(i)}$ defined as in* (2)*, for any normalizing function $\sigma(\cdot)$. Then, if $R$ is the $\lceil(|D_{\text{cal}}| + 1)(1 - \alpha)\rceil^{\ddagger}$-th smallest value of the set $\{R^{(i)} : i \in \mathcal{I}_{\text{cal}}\} \cup \{\infty\}$, the guarantee* (1) *holds with intervals $\mathcal{C}_t(X) := [-R\sigma(\widehat{Y}_t) + \widehat{Y}_t, \widehat{Y}_t + R\sigma(\widehat{Y}_t)]$, $\forall t$.*

Notice in Theorem 3.2 that the normalizing factors $\sigma(\widehat{Y}_t)$ are leveraged to scale the score $R$ into $R\sigma(\widehat{Y}_t)$ across different time points $t$. For the intervals $\mathcal{C}_t(X)$ to be bounded, we need to have $\lceil(|D_{\text{cal}}| + 1)(1 - \alpha)\rceil \leq |D_{\text{cal}}|$. We note that, while the set of time points $\mathcal{T}$ is a priori unknown, we can design a prediction band for $Y$ by deriving the intervals $\mathcal{C}_t(X)$ for all time points $t \in \{1, \ldots, T\}$, for a user-defined maximum time $T \in \mathbb{N}_+$, and interpret the intervals at time points outside $\mathcal{T}$ as interpolations among the true time points (i.e., those contained in $\mathcal{T}$). For example, in Figure 3, the conformal prediction band is guaranteed to cover the true observations (black bullets) at the corresponding (a priori unknown) time points.

**Remark 3.3.** The nonconformity scores in (2) are inspired by the conformal methods in [17, 18], which guarantee coverage for fixed-time trajectories, where the sets of time points $\mathcal{T}$ and $\mathcal{T}^{(i)}$, $i = 1, \ldots, N$, are identical and known a priori. However, in the context of biomarker trajectory prediction, observations are collected at random time points for each subject, as depicted in Figure 1. Therefore, the application of the approach from [17, 18] is hindered by the lack of prior knowledge of the time points at which observations are collected for the test subject. More details on why

---

$^{\ddagger}$The notation $\lceil \cdot \rceil$ represents the ceiling function.

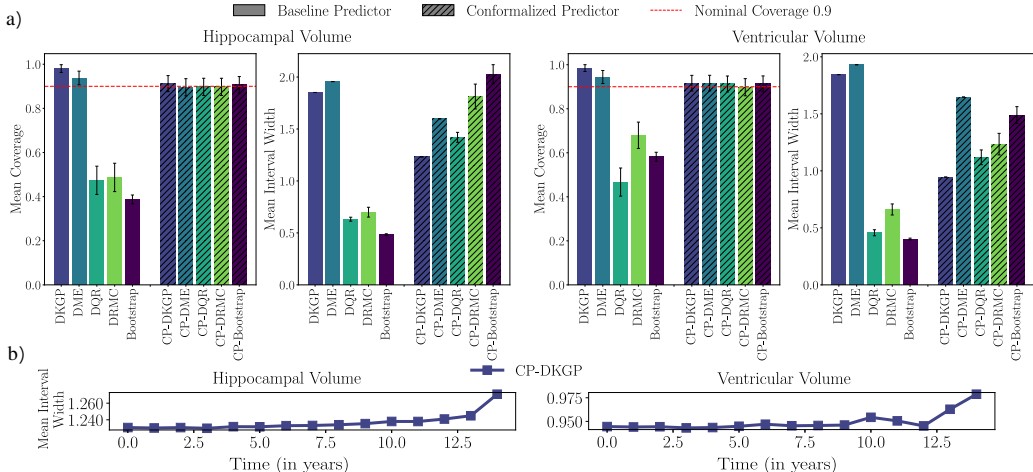

Figure 2: (a) We compare the mean coverage and mean interval width of our baseline and conformal prediction bands for hippocampal- and ventricular-volume trajectories. Error bars denote the 95th percentile of the metrics across 10 data splits. The baseline predictors (solid bars) either exceed the desired coverage by a noticeable margin (DKPG, DME) or fall short of it (DQR, DRMC, Bootstrap). In contrast, all conformalized predictors (striped bars) achieve the nominal coverage while also maintaining relatively tight intervals. (b) We show the temporal evolution of the mean interval width of the 90% conformal prediction bands for hippocampal- (left) and ventricular-volume (right) trajectories. The curve corresponds to the CP-DKGP predictor and yields relatively tight intervals early in time, with bands steadily widening as the prediction horizon extends, reflecting the expected growth in uncertainty over time. Reported interval widths are on the standardized scale.

these previous approaches cannot be used in our setting are given in F. Our method removes this assumption and guarantees coverage of randomly-timed trajectories (see Theorem 3.2).

Beyond the coverage guarantee (1), our conformal prediction method provides us with a distribution-agnostic framework for obtaining uncertainty-aware variants of arbitrary trajectory predictors. In the following section, we demonstrate the applicability of our approach to several standard and state-of-the-art predictive models.

## 4   Prediction of Brain Biomarker Trajectories

In this section, we apply our conformal prediction method in a case study involving two well-established brain biomarkers. Particularly, we focus on predicting hippocampal- and ventricular-volume trajectories, both of which are critical for diagnosing and monitoring the progression of neurodegenerative diseases, such as Alzheimer's disease (AD) [31]. For each biomarker, we use a dataset of $2,200$ samples with $(X^{(i)}, \mathcal{T}^{(i)}, Y^{(i)})$ from subject $i$: i) $X^{(i)}$ are input features consisting of demographic and clinical covariates, such as sex, race, and clinical diagnosis, as well as volume measures from 145 brain regions, collected on the subject's first visit; ii) $\mathcal{T}^{(i)}$ includes the set of time points of the subject's subsequent visits, relative to their first visit; iii) $Y^{(i)}$ is the output trajectory involving the corresponding biomarker values at the observed time points. For our experiment, we combine the preprocessed and harmonized neuroimaging measures from two well-known longitudinal studies— Alzheimer's Disease Neuroimaging Initiative (ADNI) ADNI [32] and Baltimore Longitudinal Study of Aging (BLSA) [33]—which focus on AD and Brain Aging, respectively. Details on the studies and our preprocessing pipelines can be found in Appendix D.

To demonstrate the model-agnostic nature of our framework, we apply it to various standard and state-of-the-art predictors: i) Deep Kernel Gaussian Process (DKGP) model [24], ii) Deep Mixed Effects (DME) model [20], iii) Deep Quantile Regression (DQR) model [34], iv) Deep Regression with Monte Carlo (DRMC) dropout model [35], and v) Bootstrap Deep Regression model [36]. Details on the architecture and training of each model can be referred to Appendix E.1. We note that

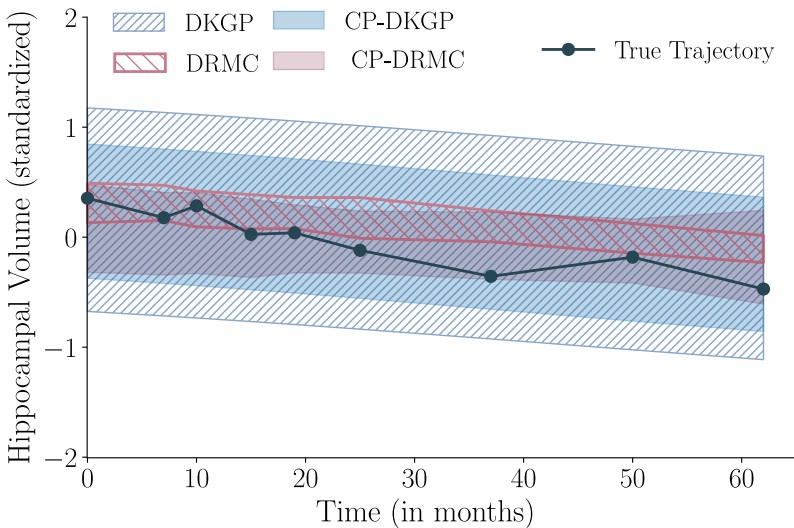

Figure 3: We compare trajectory bounds for DKGP and DRMC before and after conformal calibration. The baseline DKGP model produces overly wide bounds, while DRMC yields narrow, overconfident intervals that fail to capture later observations. After conformalization, CP–DKGP and CP–DRMC adjust their bounds to achieve the nominal coverage, correcting over- and under-confidence respectively.

existing conformal prediction methods are designed for fixed-time settings and cannot be applied to randomly-timed trajectories, which precludes direct comparisons. We elaborate further in Appendix F.

For each predictor, we present a comparative case study between its *baseline* variant, which is trained on the entire given dataset, and its *conformalized* variant, which is trained on a subset of the given dataset, as described in Subsection 3.2. To design prediction bands for the baseline variants, we employ the inherent uncertainty measure provided by their predictive standard deviation (see Appendix E.2 for details). By leveraging the same uncertainty measure, we define the normalizing function $\sigma(\cdot)$ employed in the design of our conformal prediction bands (see Theorem 3.2). We note that while baseline bands provide a notion of uncertainty-aware prediction, the desired coverage might not be achieved when model assumptions are violated.

In all experiments, we evaluate the performance of the baseline and conformal prediction bands in terms of mean coverage (i.e., the proportion of test trajectories entirely contained within the prediction bands) and mean interval width (across all test trajectories and corresponding time points). Specifically, we compute these metrics by averaging over 10 random splits of the dataset into given data and test data, with the test set comprising 10% of the total dataset. For the design of our conformal prediction bands, we perform an additional splitting of the given data into training and calibration sets. In all cases, the calibration set size is selected as 20% of the given dataset (see Appendix G.1 for details about this choice). Moreover, in this section, we consider a confidence level of 0.9 (i.e., $\alpha = 0.1$), while additional confidence levels are explored in Appendix G.2.

In Figure 2a), we illustrate the mean coverage and mean interval width attained by all the baseline and conformalized predictors for the two biomarkers of interest. We observe that in the case of the DKGP and DME predictors, both the baseline and conformalized variants achieve the desired coverage. However, the mean coverage of the conformalized predictors is closer to 0.9, and the corresponding prediction bands are tighter, as indicated by their reduced mean interval width compared to the baselines. We also see that unlike the corresponding baseline variants, the conformalized DQR, DRMC, and Bootstrap predictors are able to achieve the desired coverage, confirming our coverage guarantee in (1). Figure 2b shows the temporal evolution of conformal interval width for CP–DKGP. The intervals remain stable over time and increase only gradually with prediction horizon. The mean interval width with time for the remaining predictors are provided in Appendix G.3. To illustrate the bands qualitatively, we visualize baseline and conformal prediction intervals for hippocampal-volume trajectories from DKGP and DRMC on an example subject in Figure 3.

# 5 Group-Conditional Application across Covariate Subpopulations

So far, we have developed and tested conformalized predictors of biomarker trajectories that guarantee coverage across the *overall patient population*. However, these guarantees may not hold uniformly when applied to *subpopulations* defined by demographic or clinical covariates. In particular, high-risk groups (e.g., individuals with MCI or genetic risk factors) and underrepresented demographic subgroups may be inadequately captured by population-based conformal bands, leading to miscalibrated uncertainty estimates. To address this, we employ a group-conditional variant of our approach, directly derived from the conformal framework in [9, 37]. We describe this method in Subsection 5.1 and evaluate it in our case study on hippocampal- and ventricular-volume trajectories in Subsection 5.2.

## 5.1 Group-Conditional Conformal Prediction for Randomly-Timed Trajectories

In this subsection, we present a group-conditional variant of the conformal prediction method described in Subsection 3.2. Specifically, we provide group-conditional conformal prediction bands for randomly-timed trajectories, following the Mondrian conformal prediction framework from [9, 37].

Mondrian conformal prediction is a general approach for deriving group-conditional variants of conformalized predictors. To apply it to our setting of randomly-timed trajectories for distinct covariate subpopulations, consider a grouping function $G : \mathbb{R}^d \to \mathcal{G}$, where $\mathcal{G}$ is a predefined set of covariate groups. The idea is to stratify the calibration data by group, and apply the conformal method from Subsection 3.2 to each of the groups. In this way, we can derive group-conditional conformal bands that guarantee coverage within each covariate subpopulation. We formalize our result in the following corollary, adopting the notation from Section 3. The proof is omitted, as it directly follows from the application of Theorem 3.2 across all subpopulations, similar to the proof of [37, Proposition 3].

**Corollary 5.1** (Group-Conditional Conformal Prediction Bands for Randomly-Timed Trajectories).
*Fix a failure probability $\alpha \in (0, 1)$. Let $\widehat{Y}_t$ be the prediction of the future observation $Y_t$ at time point $t$. Consider the nonconformity scores $R^{(i)}$ defined as in (2), for any normalizing function $\sigma(\cdot)$. For each $g \in \mathcal{G}$, let $D_{\mathrm{cal},g} := \{(X^{(i)}, \mathcal{T}^{(i)}, Y^{(i)}) \in D_{\mathrm{cal}} : G(X^{(i)}) = g\}$ be the corresponding subset of $D_{\mathrm{cal}}$ and $\mathcal{I}_{\mathrm{cal},g} := \{i \in \mathcal{I}_{\mathrm{cal}} : G(X^{(i)}) = g\}$ the corresponding set of indices. Moreover, for each $g \in \mathcal{G}$, let $R_g$ denote the $\lceil(|D_{\mathrm{cal},g}| + 1)(1 - \alpha)\rceil^{\ddagger}$-th smallest value of the set $\{R^{(i)} : i \in \mathcal{I}_{\mathrm{cal},g}\} \cup \{\infty\}$. Then, for every $g \in \mathcal{G}$, we have:*

$$\mathbf{P}\left(\forall t \in \mathcal{T} : Y_t \in \mathcal{C}_t(X) \,|\, G(X) = g\right) \geq 1 - \alpha,$$

*with $C_t(X) = [-R_{G(X)}\sigma(\widehat{Y}_t) + \widehat{Y}_t, \widehat{Y}_t + R_{G(X)}\sigma(\widehat{Y}_t)], \forall t$.*

Simply put, the above corollary guarantees that, given a test example from a known covariate group $g$, we can derive conformal prediction intervals with guaranteed coverage of $1 - \alpha$ for the unknown trajectory $Y$, for any $\alpha \in (0, 1)$.

## 5.2 Application across Heterogeneous Demographic and Clinical Covariate Groups

In this subsection, we test our group-conditional conformal prediction method in the case study of hippocampal- and ventricular-volume trajectories, described in Section 4. Specifically, we design group-conditional conformal prediction bands for a total of five population stratifications, each based on a distinct demographic or clinical covariate. Among demographic factors, we consider sex, race, and education level, with the following respective subpopulations: i) females and males, ii) Asians, Blacks, and Whites, and iii) subjects with less than 16 years of education (Edu $< 16$) and more than 16 years of education (Edu $> 16$). As for clinical covariates, we consider diagnosis and APOE4 allele status, a genetic risk factor associated with Alzheimer's disease. As noted in Section 4, our diagnostic composition divides subjects into: Cognitively Normal (CN), Mildly Cognitively Impaired (MCI), and subjects diagnosed with Alzheimer's disease (AD). Moreover, subjects are classified based on their APOE4 status as non-carriers, heterozygotes and homozygotes depending on whether they have zero, one, or two copies of the APOE4 allele, respectively.

To demonstrate the impact of our group-conditional conformal method, we compare it with the approach from Subsection 3.2 for the five population stratifications. Herein, we refer to the latter method as *population conformal prediction*, owing to its coverage guarantees across the overall population. Our experiments focus on the setup of Section 4, employing the state-of-the-art Deep Kernel Gaussian Process model from [24], as well as the combined cohort of the ADNI and BLSA

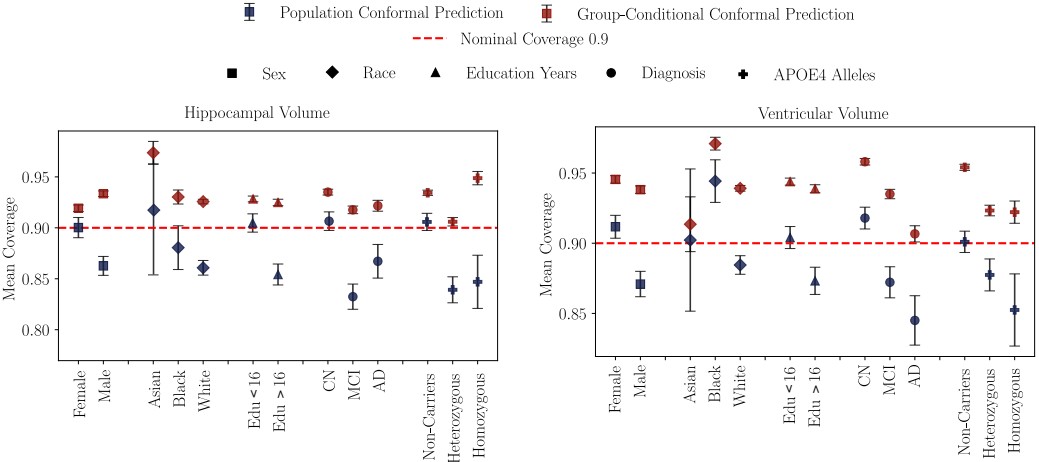

Figure 4: We compare the mean coverage of *population* and *group-conditional* conformal prediction bands for hippocampal- and ventricular-volume trajectories. Error bars denote the 95th percentile of the metrics across 10 data splits. Results are presented across five population stratifications based on individual covariates. For most covariate groups, the population conformalized predictors fail to achieve the desired coverage, while the group-conditional conformalized predictors consistently attain the desired confidence level.

studies. For the composition of the dataset across all five population strata, we refer the readers to Table 3 in Appendix D. We conduct experiments for 10 distinct splits of the data into training, calibration, and test sets, as detailed in Section 4. For each of the splits, we first learn a model from the corresponding training data, and then design: i) a single population conformalized predictor, and ii) five group-conditional conformalized predictors for the respective population stratifications. For the group-conditional conformal prediction bands, the test data are stratified by group. The mean coverage and mean interval width are computed by averaging over all 10 data splits. Results are presented for a confidence level of 0.9, while additional values are in Appendix H.

Figure 4 shows the mean coverage achieved by population- and group-conditional conformal predictors across five covariate stratifications for hippocampal-volume trajectories. We observe that the population conformalized predictors (in blue) fail to achieve the nominal coverage across most covariate groups in several cases—including clinically high-risk or underrepresented groups such as individuals diagnosed with Mild Cognitive Impairment (MCI), APOE4 homozygotes, and Asian subjects. On the other hand, while group-conditional predictors (red) consistently attain the nominal 90% coverage across all subgroups. These results confirm that group-conditional conformal bands provide more reliable coverage across diverse patient subpopulations and underscore the translational potential of our approach in clinical settings where early risk identification is essential. In the next section, we illustrate how conformal uncertainty can be directly leveraged in a downstream clinical task, enabling more informed decision-making and improving individual-level risk stratification.

## 6 Clinical Utility of Conformal Bands: Identifying High-Risk Subjects

Having established coverage guarantees, we now focus on clinical utility for identifying high-risk individuals. Specifically, we apply our conformal bounds to stratify MCI patients by risk of Alzheimer's progression, enabling early detection and targeted intervention. In clinical practice, the rate of change (RoC) of a longitudinal biomarker between the subject's first clinical visit and a subsequent clinical visit has been widely used as a progression marker for early detection and cohort enrichment [2, 38–40]. In the case of predicted biomarker trajectories, we can define the *predicted rate of change* $\widehat{\text{RoC}}$ of subject $i$ as the slope of the predicted trajectory between an initial and a future time point, denoted by $t_0$ and $t_N$, respectively, as follows:

$$\widehat{\text{RoC}}^{(i)} \;=\; \frac{\widehat{Y}_{t_N}^{(i)} - Y_{t_0}^{(i)}}{t_N - t_0}. \tag{3}$$

While the predicted $\widehat{\text{RoC}}$ is widely used as a progression marker, relying on trajectory predictions alone can underestimate risk in uncertain cases. We therefore introduce the *Rate of Change Bound*

(RoCB) as an uncertainty-aware progression metric that quantifies the worst-case rate of change with high probability. Let $L_t^{(i)}$ and $U_t^{(i)}$ denote the lower and upper bounds of an uncertainty interval obtained for subject $i$ at time point $t$ with a given method (e.g., a baseline method or our conformal prediction method). To account for the different behaviors of distinct biomarkers, the RoCB of subject $i$ is defined as follows:

$$
\text{RoCB}^{(i)} = \begin{cases} \dfrac{L^{(i)}(t_N) - Y^{(i)}(t_0)}{t_N - t_0}, & \text{if biomarker decreases with progression,} \\[2mm] \dfrac{U^{(i)}(t_N) - Y^{(i)}(t_0)}{t_N - t_0}, & \text{if biomarker increases with progression.} \end{cases} \tag{4}
$$

Note that the RoCB provides a notion of an uncertainty-aware slope. In particular, for decreasing biomarkers, such as hippocampal volume, the worst-case decline is captured by the lower bound of the interval, whereas for increasing biomarkers, such as ventricular volume, the upper bound of the interval is used. In the experiments below, we focus on hippocampal volume and therefore compute RoCB using the lower bound.

In this experiment, we focus on the hippocampal-volume trajectory as a key neurodegenerative biomarker. We evaluate each subject's predicted rate of change as a baseline progression metric and compare it to the proposed rate of change bound. The $\widehat{\text{RoC}}$ is defined as the slope between the baseline prediction and the future predicted mean value, while the RoCB is computed using the lower bound of the prediction interval at the final timepoint.

Table 1: Youden-optimised discrimination on $z$-standardised predicted rate of change $\widehat{\text{RoC}}$ and rate of change bound RoCB for MCI converters, with 95% bootstrap confidence intervals. For hippocampal volume (a decreasing biomarker), RoCB corresponds to the lower bound of the interval.

| Method | Metric | $\tau^{\star}$ | Precision | Recall | $\mathbf{F_1}$ |
|--------|--------|------|-----------|--------|------|
| DRMC | $\widehat{\text{RoC}}$ | $-0.006$ | $\mathbf{0.436 \pm 0.022}$ | $0.671 \pm 0.058$ | $0.528 \pm 0.023$ |
| | RoCB | $-0.012$ | $0.403 \pm 0.022$ | $\mathbf{0.884 \pm 0.058}$ | $\mathbf{0.553 \pm 0.023}$ |
| CP–DRMC | $\widehat{\text{RoC}}$ | $-0.006$ | $\mathbf{0.432 \pm 0.022}$ | $0.740 \pm 0.095$ | $0.546 \pm 0.024$ |
| | RoCB | $-0.020$ | $0.395 \pm 0.022$ | $\mathbf{0.915 \pm 0.095}$ | $\mathbf{0.552 \pm 0.024}$ |

We analyze subjects diagnosed with MCI at their first clinical visit. Those remaining MCI at all follow-ups are labeled MCI-Stable; those converting to Alzheimer's are MCI-Progressors. A threshold $\tau$ applied to RoC or RoCB determines trial eligibility: subjects below $\tau$ are classified as high-risk. The cohort includes 462 MCI-Stable participants (mean follow-up 22.5 months; range 0–153) and 258 MCI-Progressors (mean 23.5 months; range 0–142).

To evaluate the clinical value of our conformal bands, we compare the discriminative performance of DRMC (with its baseline uncertainty measures—see Section 4) and its conformalized counterpart (CP–DRMC) using both the predicted rate of change and rate of change bound. For each metric (RoC or RoCB), we determine the optimal threshold $\tau^*$ via Youden's index. As shown in Table 1, the RoCB metric substantially improves recall within both models. For DRMC, recall increases from 67.1% ($\widehat{\text{RoC}}$) to 88.4% (RoCB), and for CP–DRMC, from 74.0% to 91.5%. These increases demonstrate the benefit of incorporating predictive uncertainty into the slope-based progression metric. Importantly, even under the same RoCB metric, conformalization yields an additional improvement: CP–DRMC achieves a recall of 91.5% compared to 88.4% for DRMC. This increase highlights the impact of conformal calibration itself: it not only produces well-calibrated prediction intervals, but it also systematically improves risk identification by lowering uncertainty bounds. Although precision decreases modestly when using RoCB instead of RoC, the $F_1$ score remains similar, confirming that the increased sensitivity does not come at the cost of overall discrimination. Appendix Section I (Table 4) presents the quantitative results for the remaining predictors.

To further illustrate how conformal prediction improves clinical decision-making, Figure 5 presents two representative hippocampal-volume trajectories comparing DRMC and CP–DRMC. On the left, an MCI subject who converts to AD is misclassified by DRMC: the model's overconfident prediction band fails to capture the true trajectory, and the resulting RoCB ($-0.011\,\text{SD month}^{-1}$) lies above the Youden-optimal threshold ($\tau^{\star}$), leading to a missed detection. In contrast, CP–DRMC produces a wider and better-calibrated prediction band that lowers the RoCB to ($-0.020\,\text{SD month}^{-1}$), correctly classifying the subject as an AD progressor. On the right, we show a stable MCI subject incorrectly

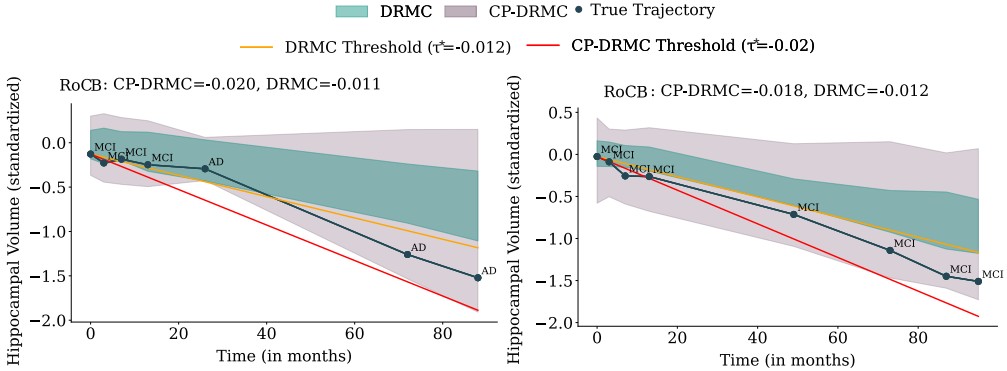

Figure 5: We visualize hippocampal-volume trajectories for two MCI participants. On the left, a converter from MCI to AD. The over-confident DRMC band (green) fails to cover the true trajectory and yields a lower-bound rate of change of $-0.011$,mo$^{-1}$—above the Youden-optimal threshold—so the subject would be missed. Conformal calibration (CP-DRMC, lilac) lowers the band, produces $\mathrm{RoCB} = -0.020$,mo$^{-1} \leq \tau^{\star}$, and correctly flags the converter. On the right, a non-converter who remains MCI. DRMC erroneously predicts conversion, whereas the wider CP-DRMC band raises the $\mathrm{RoCB}$ above the threshold and prevents a false enrolment. Together these examples *demonstrate* how conformalisation rescues under-scored converters while simultaneously reducing false positives among stable cases.

identified as high-risk by DRMC due to an overly steep $\widehat{\mathrm{RoC}}$. The conformal predictor expands the interval, raising the $\mathrm{RoCB}$ to a larger value and avoiding a false inclusion.

# 7 Discussion on Clinical Impact

Our framework brings distribution-free, uncertainty-calibrated forecasting to biomarker trajectories observed at *randomly-timed* visits—a pervasive reality in longitudinal clinical studies. By conformalizing trajectory predictions, we equip downstream decisions (e.g., early intervention, enrichment for MCI or APOE4 subgroups) with formal coverage guarantees at subject-specific time points. Importantly, our approach is model-agnostic and can conformalize any biomarker predictor to enhance reliability in precision trial design [41], where slope-based metrics guide inclusion to the clinical trial [2]. Our rate of change bound ($\mathrm{RoCB}$) uses the conformal band to favor conservative, safety-oriented screening without parametric uncertainty assumptions. Both population- and group-conditional calibration improve trustworthiness in high-risk or underrepresented subgroups.

# 8 Limitations and Future Work

Our contribution adapts split conformal and group-conditional conformal prediction to randomly-time settings via a time-varying normalizing function. We focus on *single* biomarkers; joint guarantees across coupled trajectories (e.g., hippocampus, ventricles, cognitive measures) are not yet addressed, though *multivariate* conformal prediction that provides joint coverage is a natural extension of our approach to be pursued next. Furthermore, the group-conditional conformal prediction faces challenges: as we create intersectional subgroups (e.g., APOE4 status, race, sex, age), the resulting ones become too small for effective calibration, yielding wide or uninformative bands. Future work will explore alternative approaches for multi-covariate conditioning that maintain coverage guarantees while avoiding the sample size limitations of fully stratified calibration. Additionally, our coverage guarantees assume the test and calibration data are drawn from the same underlying distribution (i.e., they are exchangeable). When this assumption is violated—such as when applying the method to new cohorts from different sites, imaging protocols, or demographic compositions—the coverage guarantees may no longer hold. Addressing distribution shift remains an open challenge. Furthermore, online variants that update prediction bands as new patient visits arrive could support continual adaptation, though the theoretical properties of such approaches in our setting remain to be established.

## Acknowledgement

This research is supported by the NIH U24NS130411 RF1AG054409 grant, the NIA contract ZIA-AG000191 and ASSET (AI-Enabled Systems: Safe, Explainable and Trustworthy) Center.

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

## Technical Appendices

## A   Exchangeability

Exchangeability of a sequence of random variables implies that their joint probability distribution remains unchanged under any permutation of the variables. This condition is milder than the standard one of independent and identically distributed random variables. Below we provide the formal definition of exchangeability from [10].

**Definition A.1.** The variables $Z^{(1)}, \ldots, Z^{(N+1)}$ are exchangeable if for every permutation $\pi(\cdot)$ of the integers $1, \ldots, N+1$, the variables $W^{(1)}, \ldots, W^{(N+1)}$, where $W^{(i)} = Z^{(\pi(i))}$, have the same joint probability distribution as $Z^{(1)}, \ldots, Z^{(N+1)}$.

## B   Proof of Theorem 3.2

For simplicity of presentation, we introduce the notation $(X^{(N+1)}, \mathcal{T}^{(N+1)}, Y^{(N+1)})$ to represent the test sample $(X, \mathcal{T}, Y)$ and the notation $R^{(N+1)}$ to represent the random variable $R$ defined in the theorem statement. Since the random variables $(X^{(i)}, \mathcal{T}^{(i)}, Y^{(i)})$, $i \in \{1, \ldots, N+1\}$, are exchangeable by assumption, the random variables $(X^{(i)}, \mathcal{T}^{(i)}, Y^{(i)})$, $i \in \mathcal{I}_{\text{cal}} \cup \{N+1\}$, are also exchangeable. Somewhat abusing notation, we define the nonconformity score function:

$$R(X, \mathcal{T}, Y) = \max_{t \in \mathcal{T}} \left\{ \frac{|Y_t - \widehat{Y}_t|}{\sigma(\widehat{Y}_t)} \right\},$$

that applied to each datapoint $(X^{(i)}, \mathcal{T}^{(i)}, Y^{(i)})$, $i \in \mathcal{I}_{\text{cal}} \cup \{N+1\}$, yields the corresponding score $R^{(i)}$. Conditioned on the dataset $D_{\text{train}}$, the function $R(\cdot)$ is deterministic. Therefore, we conclude that conditioned on $D_{\text{train}}$, the random variables $R^{(i)}$, $i \in \mathcal{I}_{\text{cal}} \cup \{N+1\}$, are exchangeable. Hence, from [42, Lemma 1] we obtain the conditional property:

$$\mathbf{P}\left( \max_{t \in \mathcal{T}} \left\{ \frac{|Y_t - \widehat{Y}_t|}{\sigma(\widehat{Y}_t)} \right\} \leq R \,\Big|\, D_{\text{train}} \right) \geq 1 - \alpha, \tag{5}$$

where $R$ is defined as in the theorem statement. By marginalizing over $D_{\text{train}}$, (5) yields the unconditional property:

$$\mathbf{P}\left( \max_{t \in \mathcal{T}} \left\{ \frac{|Y_t - \widehat{Y}_t|}{\sigma(\widehat{Y}_t)} \right\} \leq R \right) \geq 1 - \alpha. \tag{6}$$

The guarantee (1) directly follows from rewriting (6) as:

$$\mathbf{P}\left( \forall t \in \mathcal{T} : |Y_t - \widehat{Y}_t| \leq R\sigma(\widehat{Y}_t) \right) \geq 1 - \alpha$$

and setting $\mathcal{C}_t(X) = [-R\sigma(\widehat{Y}_t) + \widehat{Y}_t, \widehat{Y}_t + R\sigma(\widehat{Y}_t)]$, for all $t$.

∎

## C   Discussion on Clinical Impact

Our conformal prediction framework provides prediction bands that enable clinicians to perform informed prognosis and decision-making with greater reliability. This is important as predictive models are increasingly applied in healthcare for both patient management and drug development. In the latter case, Cummings et al. [41] highlighted the need for AI-informed clinical trials, referred to as precision trial design. Along these lines, Maheux et al. [2] evaluates a predictive model for biomarker trajectories, in the context of AD, where derived measures—such as biomarker rate of change—serve as quantitative indicators of whether a subject is likely to progress to the disease during the clinical trial. This assessment ultimately informs decisions on subject inclusion in the drug administration process. By conformalizing such predictors, either population level or calibrated in covariate groups, such as MCI subjects, we provide confidence bands alongside predictions, thus increasing the reliability of these critical clinical decisions.

Table 2: Demographic and clinical characteristics of the clinical studies. The joint cohort of ADNI and BLSA comprises our base population, whereas the remaining cohorts represent external populations. For the total time of observation and the age, we report the mean and standard deviation over all subjects contained in each study. CN: Cognitively Normal, MCI: Mild Cognitive Impairment, AD: Alzheimer's Disease.

| Study | Subjects | Obs. Time (mo) | Males (%) | Age | Diagnosis (%) | | |
| | | | | | CN | MCI | AD |
| --- | --- | --- | --- | --- | --- | --- | --- |
| ADNI+BLSA | 2200 | 65 ± 39 | 53.2 | 75.4 ± 8.6 | 49.3 | 34.8 | 15.9 |

Table 3: Composition of the joint cohort of the ADNI and BLSA studies across five demographic and clinical covariates.

| Covariate | Groups | Percentage of Subjects | Number of Subjects |
| --- | --- | --- | --- |
| **Sex** | Male | 52.89 | 1164 |
| **Race** | White | 87.46 | 1924 |
| | Black | 8.60 | 189 |
| | Asian | 3.94 | 87 |
| **Education Years** | Less than 16 | 54.56 | 1200 |
| | More than 16 | 45.44 | 1000 |
| **Diagnosis** | Cognitively Normal | 49.12 | 1081 |
| | Mild Cognitive Impairment | 34.76 | 765 |
| | Alzheimer's Disease | 16.01 | 352 |
| **APOE4 Status** | Heterozygous | 32.16 | 707 |
| | Homozygous | 7.31 | 161 |
| | None | 60.54 | 1332 |

# D   Clinical Datasets and Preprocessing

Our data consists of neuroimaging and demographic measures taken from subjects in the iSTAGING consortium [43]. Specifically, the neuroimaging measures are the 145 anatomical brain Regions of Interest (ROI) volumes (119 ROIs in gray matter, 20 ROIs in white matter and 6 ROIs in ventricles) extracted using a multi-atlas label fusion method [44]. Phase-level harmonization is applied on these 145 ROI volumes to remove site effects [45]. To train, calibrate and test the conformalized predictors we use data from two cohorts: the Alzheimer's Disease Neuroimaging Initiative [32], which is a public-private collaborative longitudinal cohort study and has recruited participants categorized as Cognitively Normal (CN), Mildly Cognitively Impaired (MCI) and diagnosed with Alzheimer's Disease (AD) through 4 phases (ADNI1, ADNIGO and ADNI2) [46]. We also use Baltimore Longitudinal Study of Aging (BLSA) follows participants who are cognitively normal at enrollment with imaging and cognitive exams since 1993 [33].

We also extract from iSTAGING cohort: the OASIS [47], The Wisconsin Registry for Alzheimer's Prevention (WRAP) study [48], the Australian Imaging, Biomarker, and Lifestyle (AIBL) study [49], the Coronary Artery Risk Development in Young Adults (CARDIA) [50], the PreventAD [51] and PENN.

For the clinical variables, we utilize Age at Baseline, Sex, Years of Education, and APOE4 Allele status, the latter being a known risk factor for Alzheimer's Disease. Diagnostic categories were designated as Cognitively Normal (CN), Mild Cognitive Impairment (MCI), and Alzheimer's Disease (AD). Subjects diagnosed with alternative forms of dementia, such as Lewy Body Dementia and Frontotemporal Dementia, were excluded from the study. These exclusions are minimal (less than 10 subjects) and did not impact the overall sample size. After filtering, our dataset consists of 2200 subjects. Furthermore, Years of Education was dichotomized: subjects with more than 16 years of education were coded as '1', while those with 16 years or fewer were coded as '0'. Also, duplicate acquisitions within the same month are discarded.

# E Predictors

## E.1 Architectural Design and Training

Our input data consists of multivariate features, including volumetric imaging features, demographic information, and clinical variables. All our predictors generate biomarker trajectories by using a learned model with an additional time input variable $t \in \mathbb{N}_+$, that outputs an estimate of the corresponding future biomarker value $\widehat{Y}_t$. Below we present five examples of such models, along with corresponding learning algorithms.

**Deep Kernel Regression.** A fully connected feedforward neural network is used to linearly transform input data into a low-dimensional latent space. The transformed input is then passed to a Gaussian Process with a zero mean function and an radial basis function (RBF) kernel. The GP component is trained using exact inference by minimizing the negative marginal log-likelihood. For details on exact GP inference, refer to [19]. DKGP approach builds upon the deep kernel learning paradigm presented in [52] and applied for trajectory prediction in [24]. The DKGP is trained for 100 epochs, using the Adam optimizer [53] with weight decay 0.02 and a learning rate of 0.01 for the deep network parameters and 0.2 for the hyperparameters of the Gaussian Process.

**Deep Mixed Effects.** The DME model leverages an embedding network and a deep mean function to capture both global trends and local variations in structured regression tasks. The embedding network projects inputs into a 64-dimensional latent space, while the mean function, implemented as an MLP, maps the latent features to a scalar regression output. The Gaussian Process (GP) with an RBF kernel operates on the latent space to model residuals, with warping applied when an embedding function is used. The model is trained end-to-end using variational inference, minimizing the negative Evidence Lower Bound (ELBO). Separate Adam optimizers are employed for the mean function and GP kernel parameters, each using a learning rate of $10^{-3}$ and an $L_2$-penalty of $10^{-3}$. For each training iteration, the GP parameters are adapted over $n_{\text{adapt}} = 10$ steps with an inner learning rate of $10^{-2}$. The training process spans 50 epochs, alternating between optimizing the mean function and GP parameters. Details on the DME model can be found in [20].

**Deep Quantile Regression.** The model is a fully connected feedforward neural network designed for quantile regression, predicting multiple quantiles simultaneously (e.g., 0.1, 0.5, 0.9). It includes a 128-unit hidden layer and a 64-unit hidden layer, both followed by ReLU activations and dropout with a 0.2 rate for regularization. The output layer has one unit per quantile to estimate. The model is trained using a quantile loss function [54]. DQR is using Adam for the optimization, with a learning rate of 0.01 for 200 epochs.

**Deep Regression with Monte Carlo Dropout.** The model is a fully connected feedforward neural network with Monte Carlo Dropout for uncertainty estimation in regression tasks. It consists of a 128-unit hidden layer, a 64-unit hidden layer, and a single-unit output layer for continuous scalar prediction. ReLU activations are applied to the hidden layers, and a dropout layer with a fixed rate 0.2 is applied during both training and inference to approximate Bayesian uncertainty. DRMC is using Adam for the optimization with learning rate of 0.01 for 200 epochs.

**Bootstrap.** The model is a fully connected feedforward neural network for regression tasks, comprising a 128-unit hidden layer, a 64-unit hidden layer, and a single-unit output layer. ReLU activations are applied to hidden layers, and the output layer produces a raw scalar value for regression. The model is deployed as part of a bootstrap ensemble, where 10 instances are trained on different bootstrap-resampled subsets of the train data. Each instance is trained with consistent hyperparameters (epochs and learning rate) to enhance robustness and reduce variance by aggregating predictions.

## E.2 Predictive Standard Deviation

We outline the models and their uncertainty quantification mechanisms. Predictions are denoted as $\widehat{Y}_t$ with standard deviation $\sigma(\widehat{Y}_t)$.

**Deep Kernel Regression (DKGP) and Deep Mixed Effects (DME).** Both models estimate the posterior predictive distribution using exact inference. The posterior predictive distribution is a gaussian distribution, and from that we extract the predictive mean that corresponds to the point estimates $\widehat{Y}_t$ and the predictive variance that corresponds to a requested confidence level, i.e 0.90.

From that we calculate the standard deviation $\sigma(\widehat{Y}_t)$. Details on the mathematical formula of the predictive mean and predictive variance can be found in [19]

**Deep Quantile Regression (DQR).** Deep Quantile Regression predicts the desired quantiles—lower, mean, and upper—for a specific confidence level. From these predictions, we calculate the quantiles corresponding to different percentiles, such as the 10th, 50th, and 90th percentiles, enabling uncertainty quantification. The confidence level represents the range between the lower and upper percentiles. The predicted variance is estimated using the spread between the upper and lower quantiles, adjusted by a z-score corresponding to the desired confidence level (e.g., 1.645 for a 90% confidence interval). This calculation assumes that the predictive distribution is approximately Gaussian, allowing the z-score to be used as a scaling factor for the quantile spread.

**Deep Regression with Monte Carlo (DRMC) dropout.** Monte Carlo dropout approximates Bayesian inference via multiple stochastic forward passes with dropout. The model generates multiple predictions, and from these we extract the standard deviation of the sample. We run inference 100 times.

**Bootstrap Deep Regression (Bootstrap).** Bootstrap Deep Regression leverages an ensemble of 20 deep regression models, each trained on a different bootstrap sample, to estimate uncertainty. During inference, the ensemble produces multiple predictions for the same input, and the sample standard deviation of these predictions quantifies predictive uncertainty.

# F   Relation to the Broad Conformal Prediction Literature

Our conformal prediction method provides an extension to the setting of randomly timed trajectories, which is necessary for applying conformal prediction using real-world biomarker data. None of the previous methods, including [25] can handle randomly-timed trajectories. Below, we elaborate on why prior CP methods for fixed-time trajectories cannot be applied in our setting of randomly timed biomarker trajectories. Our simple yet effective normalization function is able to accommodate conformal inference on randomly-time trajectories. The normalizing function enables us to compute a normalized prediction error across all time steps, ensuring that no component within the maximum dominates the others in terms of scale. This function leverages the model's predictive standard deviation to measure prediction uncertainty, effectively normalizing errors in a standard way. Our conformal prediction intervals at each time point in a test trajectory are not influenced by the trajectory's length. The bound for the prediction error $|\widehat{Y}_t - Y_t|$ at each time $t$ is derived by scaling the score $R$ by the time-specific uncertainty estimate $\sigma(\widehat{Y}_t)$, yielding the value $R \cdot \sigma(\widehat{Y}_t)$. This time-varying scaling ensures that the conformal bands adapt to the uncertainty at each time point. Additionally, a true "worst-case" scenario would arise only if $\sigma(\widehat{Y}_t) = 1$ at all time points, which does not hold here.

Our extension of conformal prediction to randomly timed trajectories is necessary for real-world biomarker data. None of the previous methods [13, 18, 25, 30]can handle randomly-timed trajectories. Closest to our work are the approaches by [18] and [17], which are designed under the assumption of fixed-time trajectories—i.e., observations are made at the same pre-defined time points across the entire population. In contrast, our setting involves trajectories with observations collected at random times due to missed visits and variable scheduling typical in clinical studies. To adapt the method from [17, 18] to our setting, one could select a subset of the data corresponding to a fixed set of time points (e.g., 3, 7, 9, and 12 months) and extract only the corresponding observations from each subject. However, this leads to substantial dataset reduction. In a simple experiment following this approach, we found that only 37 trajectories with observations at all four selected time points remained out of the total dataset of 2200 trajectories. Beyond dataset reduction, this method limits the applicability of the conformal prediction bands. More specifically, since [18] and [17] define normalization factors $\sigma_t$ only at the pre-specified time points $t$ (e.g., 3, 7, 9, 12 months), their method can produce prediction intervals *only* at those times during inference. Thus, if a test subject visits the clinic at months 4, 5, and 14, the available intervals at months 3, 7, 9, and 12, provided by [18] and [17], do not allow valid conformal inference at months 4, 5, and 14. In contrast, we introduce a normalizing function $\sigma(\cdot)$ that produces predictive uncertainty estimates $\sigma(\widehat{Y}_t)$ for *any* time point $t$, even if no observations at that time are available in the calibration data. This allows our method to produce valid conformal intervals at arbitrary test-time points, in contrast to existing conformal prediction methods designed for fixed-time trajectories.

# G   Case Study: Details and Extended Results

## G.1   Calibration Set Size Selection

In this section, we outline the procedure for determining the suitable calibration set size for conformal prediction. Specifically, we vary the fraction of the training set used for calibration from $0.01$ to $0.30$, while the remaining portion is reserved for predictor training. We then evaluate the resulting conformal intervals on a held-out validation set in terms of two key metrics: (i) mean coverage and (ii) mean interval width.

This procedure is repeated for both hippocampus and ventricular volume biomarkers, at confidence levels of $0.90$, $0.95$, and $0.99$, and for all five conformal predictors. Our goal is to pick the size that achieves the desired coverage with the smallest possible interval width.

Our findings, visualized in figure 6 indicate that as the calibration set fraction increases, the empirical coverage exceeds the nominal coverage level. We identify a calibration set fraction of *0.20* as a practical choice that achieves coverage while avoiding wide prediction intervals. This calibration set fraction corresponds to a calibration set with $414$ subjects.

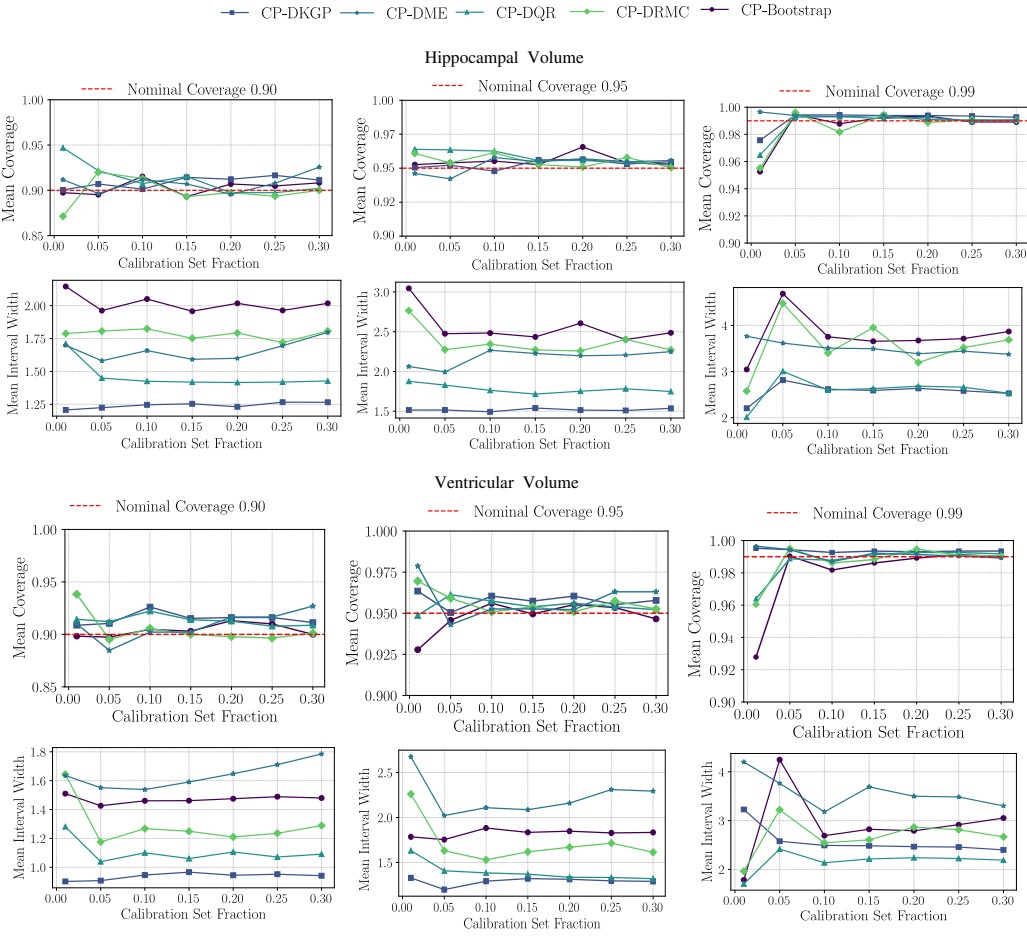

Figure 6: Mean coverage and mean interval width for hippocampal- and ventricular-volume trajectories at nominal coverage levels of *0.90*, *0.95*, and *0.99*. Error bars denote the 95th percentile of the metrics across 10 data splits. Each column corresponds to a different nominal coverage level, and each curve represents one of five conformalized predictors (CP-DKGP, CP-DME, CP-DQR, CP-DRMC, CP-Bootstrap). Dashed red lines indicate the nominal coverage. The horizontal axis shows the fraction of the training set used for calibration, ranging from *0.01* to *0.30*

## G.2 Comparison between Baseline and Conformalized Predictors for Other Confidence Levels

In this section, we provide additional results comparing the conformalized predictors with their baseline counterparts at confidence levels of 0.95 and 0.99. For both hippocampal- and ventricular-volume trajectories, we observe trends similar to those reported at the 0.90 confidence level in Section 4 of the main paper. Specifically, conformal adjustments effectively boost empirical coverage to the desired nominal level. Gaussian Process based methods, such as the DKGP and DME already achieve the nominal coverage. Conformalized-DGKP and conformalized-DME attain nominal coverage with tighter bounds, which corresponds to less conservative uncertainty quantification. On the contrary, for the methods of DQR, DRMC and Boostrap, that provide undercoverage, their conformalized versions achive nominal coverage by widening the intervals.

These findings confirm the effectiveness of our conformal prediction across varying confidence levels.

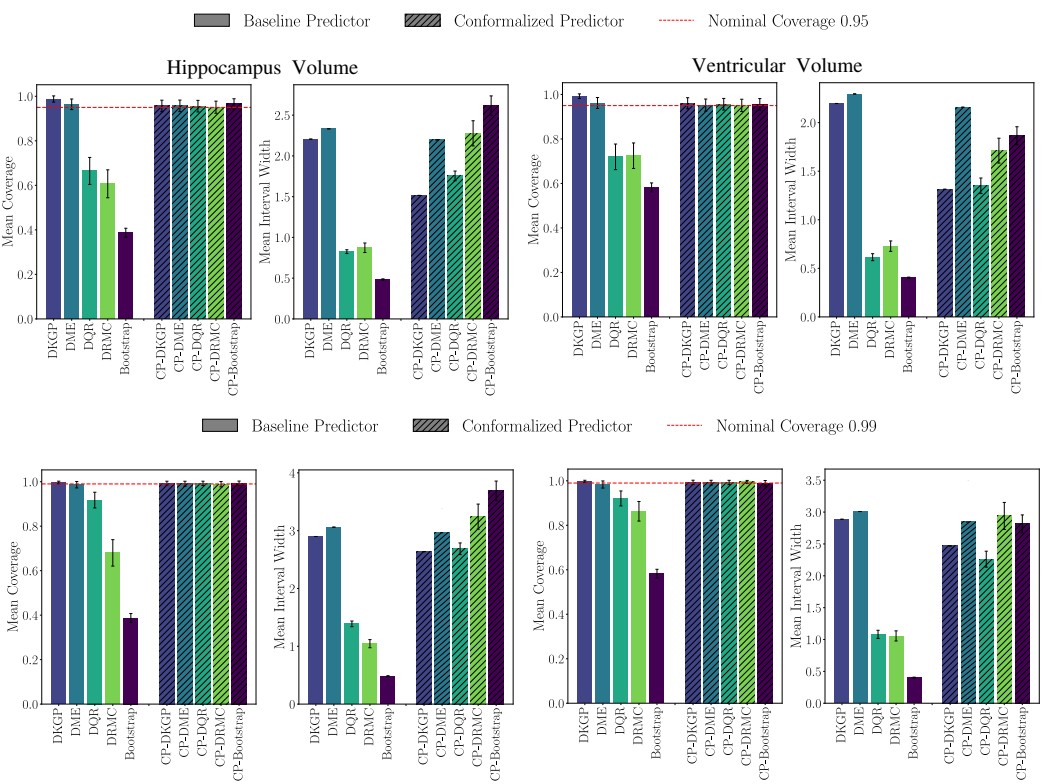

Figure 7: Mean coverage and mean interval width of baseline and conformalized predictors for hippocampal- and ventricular-volume trajectories at nominal confidence levels of 0.95 (top) and 0.99 (bottom). The dashed red line indicates the nominal coverage. Error bars denote the 95th percentile of the metrics across 10 data splits. Overall, the conformalized methods exhibit coverage closer to the nominal level, by adjusting accordingly the interval width.

Next, we demonstrate how uncertainty changes over time starting from the initial acquisition, which, in our study, corresponds to the test subject's first hospital visit.

## G.3 Mean Interval Width of the Conformalized Predictors over Time

Having established in the previous section that each conformalized baseline achieves desired coverage levels across different calibration set sizes and confidence levels, we now focus to the evolution of the conformal prediction intervals over time. In practical scenarios, the uncertainty in predicting future measurements naturally increases as we move further away from a known data acquisition, and understanding this growth in interval width is crucial for longitudinal analyses.

Figure 8 shows how the conformal prediction intervals of our selected methods vary with the time elapsed from the most recent (known) acquisition. Each row corresponds to a different conformalized predictor, while the horizontal axis indicates increasing time from the known data point, and the vertical axis shows the corresponding mean interval width. As expected, most methods exhibit relatively tight intervals when predicting only a short time ahead, but the intervals expand as the temporal distance grows. This behavior reflects the increasing uncertainty associated with predicting further into the future.

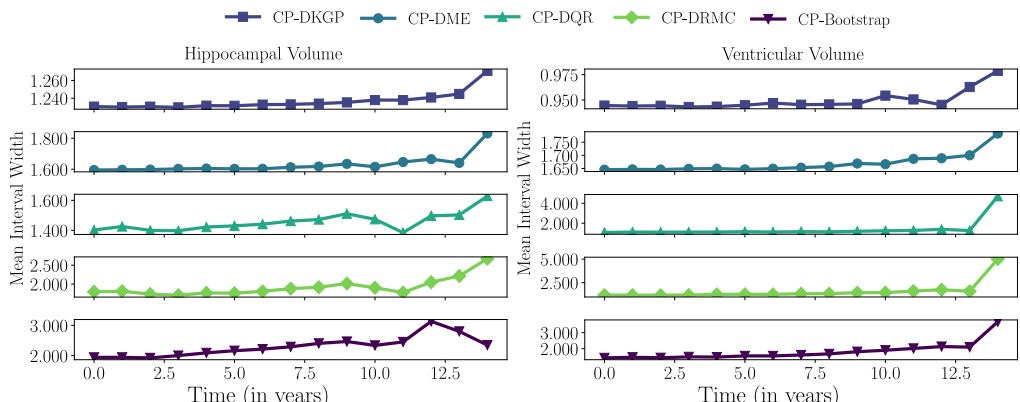

Figure 8: Temporal evolution of the mean interval width (averaged per year) for the five conformalized predictors. We observe that as time increases, the average conformal predictive intervals increase.

# H    Group-Conditional Application across Covariate Subpopulations: Extended Results

Figure 10 depicts the mean interval widths for hippocampal- and ventricular-volume trajectories across subgroups defined by demographic (Sex, Race, Education) and clinical (Diagnosis, APOE4 alleles) covariates. The population conformal prediction generates narrower intervals for all the subgroups, leading to undercoverage, particularly for high-risk or underrepresented groups such as Black and Asian participants, MCI patients, and APOE4 homozygotes. In contrast, group-conditional conformalized predictors (in red) produces wider intervals in order to ensure that empirical coverage aligns with nominal levels within the specific subpopulations. While for the hippoampal volume all the subpopulations were not covered, for the ventricular volume several subpopulations were already reaching the nominal coverage of 0.9, specifically for females, Asian, Black, subjects with less than 16 years of education as well as the Non-carriers. From the Figure 10 we observe that these are the ones that already reach the nominal level of coverage exhibit the lower increase in the interval width withing the subpopulation. For example, females have lower mean interval width than males.

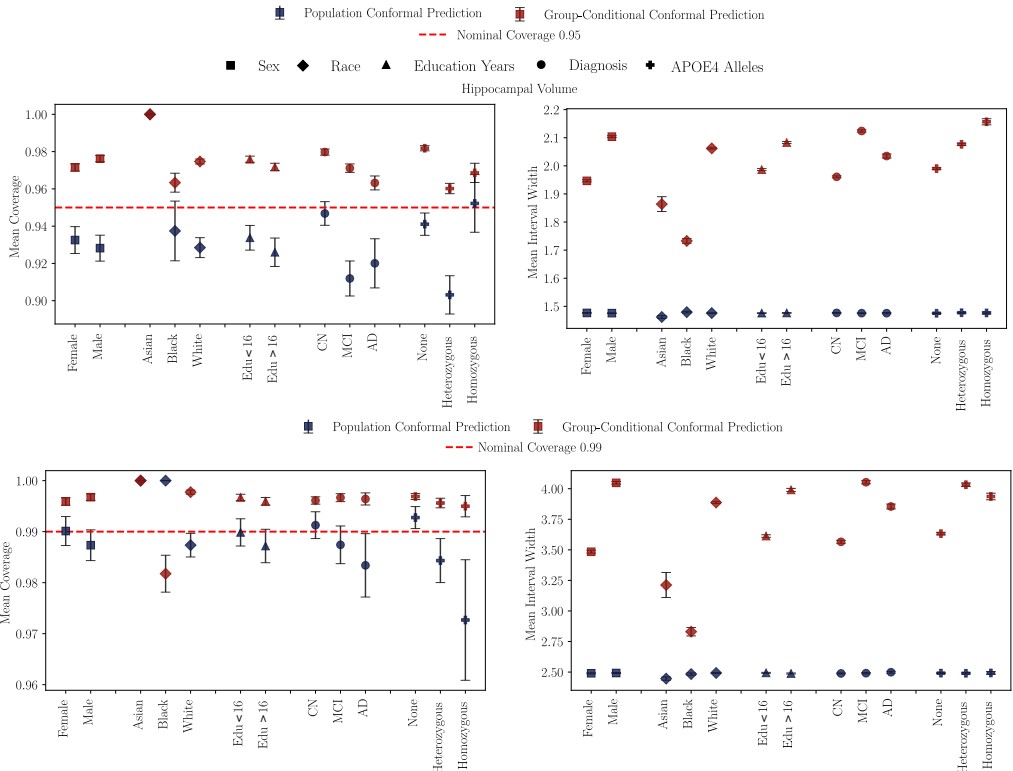

Figure 9: We compare the mean coverage of *population* and *group-conditional* conformal prediction bands for ventricular-volume trajectories. Error bars denote the 95th percentile of the metrics across 10 data splits. Results are presented across five population stratifications based on individual covariates (sex, race, education level, cognitive diagnosis, and APOE4 alleles status). We observe that *group-conditional* intervals are adjusted per covariate group in order to reach the nominal coverage. On the contrary, the *population* conformal prediction provides consistent intervals across all groups, that cause the coverage disparities.

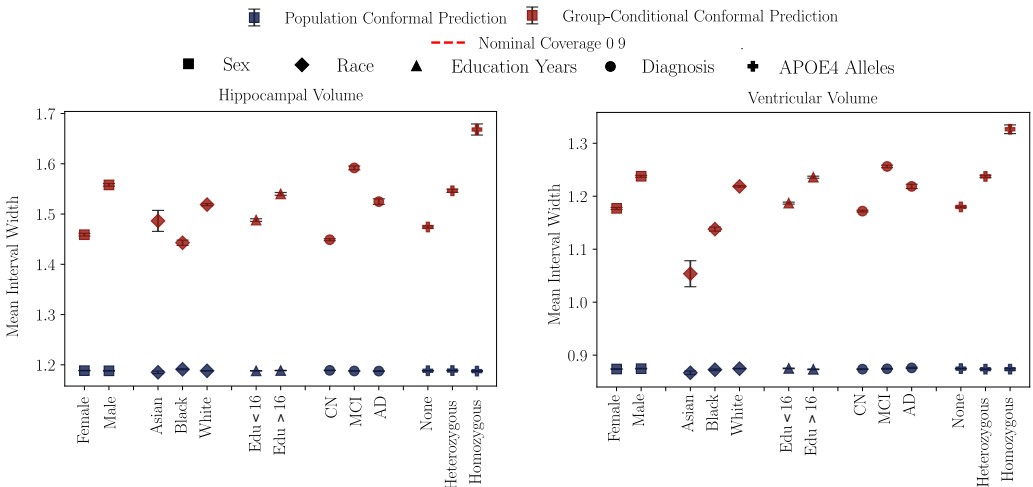

Figure 10: We compare the mean interval width of *population* and *group-conditional* conformal prediction bands for hippocampal- and ventricular-volume trajectories.Error bars denote the 95th percentile of the metrics across 10 data splits. Results are presented across five population stratifications based on individual covariates (sex, race, education level, cognitive diagnosis, and APOE4 alleles status). We observe that *group-conditional* intervals are adjusted per covariate group in order to reach the nominal coverage. On the contrary, the *population* conformal prediction provides consistent intervals across all groups, that cause the coverage disparities.

Figure 9 presents the empirical coverage and mean interval widths for hippocampal-volume trajectories across subgroups at varying confidence levels 0.95 and 0.99. Again, we observe the same trend as in the Figure 4, where population conformal prediction bands show noticeable undercoverage for underrepresented subgroups, such as Black and Asian participants, APOE4 homozygotes, and MCI patients. For the nominal level of 0.95 only the Homozygotes appear to be covered by the population conformal predictor. Again, the group-conditional conformal prediction adapts the intervals of each subpopulation in order to achieve the nominal coverage withing subpopulation. For the nominal level of 0.99 we observe that less subpopulations are miscovered and the majority of them align closely to the nominal level. This is expected as the confidence bands are wide enough in order to capture any subpopulation variability.

Figure 12 presents the empirical coverage and mean interval widths for ventricular-volume trajectories across subgroups at varying confidence levels 0.95 and 0.99. For ventricular volume conformal prediction bands, a similar pattern is observed. Population conformal prediction underperform for the subgroups defined by Diagnosis and APOE4 status. Specifically, at the nominal level of 0.95, MCI and AD subjects as well as the heterozygotes and homozygotes are not covered by the population conformal prediction. However, the group-conditional conformal prediction tackles this by widening the intervals across those subgroups and thus achieving nominal coverage.

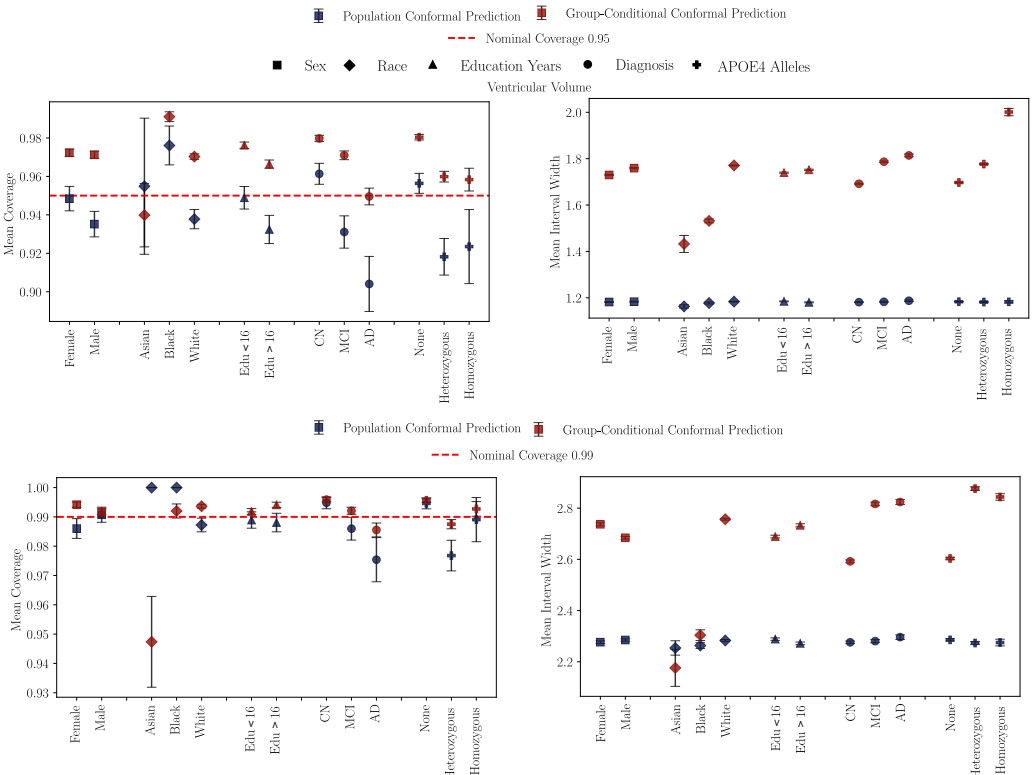

Figure 12: We compare the mean coverage and mean interval width of *population* and *group-conditional* conformal prediction bands for ventricular-volume trajectories for the nominal coverage of 0.95 and 0.99. Error bars denote the 95th percentile of the metrics across 10 data splits. Results are presented across five population stratifications based on individual covariates (sex, race, education level, cognitive diagnosis, and APOE4 alleles status).

## I  Clinical Utility of Conformal Bands: Identifying High-Risk Subjects

To complement the clinical analysis in Section 6, we further investigate the behavior of predictors that produce overconservative uncertainty estimates. Such models generate wide prediction intervals that capture most future outcomes, resulting in high recall but limited precision. While this behavior may be considered safe in clinical contexts, it can lead to over-inclusion of low-risk subjects and reduced specificity. Here, we examine whether conformalization can help refine these models by tightening the prediction bounds, thereby improving the balance between sensitivity and precision. We focus on two representative predictors: DME and DKGP. The total quantitative results are presented in Table 4.

The DME model exhibits strongly conservative behavior. Using $\widehat{\text{RoC}}$, it identifies 81.4% of MCI converters, and recall increases to 98.4% under $\text{RoCB}$, indicating near-complete inclusion of progressors. However, this recall comes at the cost of low precision—37.8% for $\widehat{\text{RoC}}$ and 36.7% for $\text{RoCB}$—resulting in many false positives. Conformalization (CP–DME) slightly tightens the intervals, leading to a modest drop in $\text{RoCB}$ recall to 94.2%, while improving precision to 36.9%.

The DKGP model also demonstrates overconservative behavior, producing wide prediction intervals that result in strong recall but imprecise discrimination. Under the $\widehat{\text{RoC}}$ metric, DKGP identifies 80.2% of MCI converters, and $\text{RoCB}$ pushes recall even higher to 92.2%. However, this gain comes with reduced precision—falling from 50.6% to 37.4%. After conformalization, CP–DKGP produces slightly narrower intervals. While this modestly lowers recall under $\widehat{\text{RoC}}$ (to 72.1%), the $\text{RoCB}$-based recall remains high at 90.3%, and precision increases slightly (to 37.6%). These results highlight the role of conformalization in refining overly broad uncertainty estimates: by reducing

excess conservativeness, CP–DKGP achieves a more balanced trade-off between identifying true progressors and limiting false positives, while still preserving the safety benefits of RoCB-based risk stratification.

Table 4: Youden-optimised discrimination on $z$-standardised rate of change (RoC) and lower-bound rate of change (RoCB) for MCI converters, with 95% bootstrap confidence intervals (CIs).

| Method | Metric | $\tau^\star$ | Precision (95% CI) | Recall (95% CI) | $F_1$ (95% CI) |
|---|---|---|---|---|---|
| DRMC | $\widehat{\text{RoC}}$ | $-0.006$ | 0.436 [0.367, 0.455] | 0.671 [0.693, 0.919] | 0.528 [0.504, 0.593] |
|  | RoCB | $-0.012$ | 0.403 [0.367, 0.455] | 0.884 [0.693, 0.919] | 0.553 [0.504, 0.593] |
| CP–DRMC | $\widehat{\text{RoC}}$ | $-0.006$ | 0.432 [0.360, 0.446] | 0.740 [0.667, 0.956] | 0.546 [0.498, 0.590] |
|  | RoCB | $-0.020$ | 0.395 [0.360, 0.446] | 0.915 [0.667, 0.956] | 0.552 [0.498, 0.590] |
| DKGP | $\widehat{\text{RoC}}$ | $-0.007$ | 0.506 [0.340, 0.414] | 0.802 [0.807, 0.993] | 0.621 [0.494, 0.573] |
|  | RoCB | $-0.019$ | 0.374 [0.340, 0.414] | 0.922 [0.807, 0.993] | 0.532 [0.494, 0.573] |
| CP–DKGP | $\widehat{\text{RoC}}$ | $-0.007$ | 0.507 [0.338, 0.414] | 0.721 [0.809, 0.996] | 0.595 [0.491, 0.573] |
|  | RoCB | $-0.015$ | 0.376 [0.338, 0.414] | 0.903 [0.809, 0.996] | 0.531 [0.491, 0.573] |
| DME | $\widehat{\text{RoC}}$ | $-0.000$ | 0.378 [0.333, 0.406] | 0.814 [0.813, 1.000] | 0.516 [0.489, 0.571] |
|  | RoCB | $-0.011$ | 0.367 [0.333, 0.406] | 0.984 [0.813, 1.000] | 0.535 [0.489, 0.571] |
| CP–DME | $\widehat{\text{RoC}}$ | $-0.000$ | 0.378 [0.335, 0.408] | 0.814 [0.836, 1.000] | 0.516 [0.492, 0.569] |
|  | RoCB | $-0.012$ | 0.370 [0.335, 0.408] | 0.942 [0.836, 1.000] | 0.531 [0.492, 0.569] |
| Bootstrap | $\widehat{\text{RoC}}$ | $-0.008$ | 0.501 [0.363, 0.450] | 0.698 [0.754, 0.930] | 0.583 [0.503, 0.588] |
|  | RoCB | $-0.012$ | 0.407 [0.363, 0.450] | 0.837 [0.754, 0.930] | 0.548 [0.503, 0.588] |
| CP–Bootstrap | $\widehat{\text{RoC}}$ | $-0.007$ | 0.454 [0.352, 0.432] | 0.733 [0.804, 0.956] | 0.561 [0.504, 0.581] |
|  | RoCB | $-0.024$ | 0.387 [0.352, 0.432] | 0.888 [0.804, 0.956] | 0.539 [0.504, 0.581] |

## I.1 Threshold-Free Evaluation of $\mathrm{RoCB}$ vs $\widehat{\mathrm{RoC}}$ Metrics

To address concerns regarding the reliance on threshold-specific performance (e.g., Youden's J), we conducted a comprehensive *threshold-free* evaluation of our uncertainty-aware biomarker, $\mathrm{RoCB}$, compared against the predicted rate of change ($\widehat{\mathrm{RoC}}$). This analysis spans ten trajectory prediction methods, including DKGP, CP-DRMC, Bootstrap, and others.

For each method, we computed ROC-AUC, PR-AUC, F1 score, precision, recall, and balanced accuracy across 720 test subjects. These metrics were chosen to reflect not only the discriminative ability (AUCs) but also the clinical decision trade-offs in high-risk detection (recall vs. precision).

$\mathrm{RoCB}$ consistently improves recall across all models, often by large margins (e.g., +62% in CP-DQR). This demonstrates its strength in identifying high-risk individuals under worst-case prediction scenarios. However, this gain often comes at the cost of reduced precision and AUC metrics, reflecting a conservative, sensitivity-focused behavior. Despite this trade-off, F1-score improves in 6 out of 10 models, with the largest increase seen in CP-DQR (+9.8%), indicating that in certain settings $\mathrm{RoCB}$ enhances both sensitivity and overall prediction balance.

These results confirm that $\mathrm{RoCB}$ is not a replacement for $\widehat{\mathrm{RoC}}$, but a *complementary metric* that prioritizes reliable early detection. This behavior is particularly beneficial in clinical contexts such as trial enrichment or preclinical screening, where high recall is often more critical than specificity.

Table 5: Threshold-free evaluation of $\widehat{\mathrm{RoC}}$ vs. $\mathrm{RoCB}$ using ROC-AUC, PR-AUC, Recall, and F1-score. Values shown as ($\widehat{\mathrm{RoC}}$ / $\mathrm{RoCB}$). LRoC improves recall consistently and maintains or improves F1 in 6 of 10 models.

| Model | AUC | PR | Rec | F1 |
|---|---|---|---|---|
| DRMC | 0.597 / 0.541 | 0.421 / 0.354 | 0.671 / 0.884 | 0.528 / 0.553 |
| CP-DRMC | 0.608 / 0.526 | 0.421 / 0.346 | 0.740 / 0.915 | 0.546 / 0.552 |
| DKGP | 0.716 / 0.435 | 0.523 / 0.302 | 0.802 / 0.922 | 0.621 / 0.532 |
| CP-DKGP | 0.697 / 0.438 | 0.502 / 0.303 | 0.721 / 0.903 | 0.595 / 0.531 |
| DQR | 0.637 / 0.508 | 0.441 / 0.334 | 0.791 / 0.841 | 0.583 / 0.539 |
| CP-DQR | 0.594 / 0.483 | 0.431 / 0.325 | 0.519 / 0.841 | **0.482 / 0.529** |
| DME | 0.495 / 0.412 | 0.370 / 0.293 | 0.814 / 0.984 | 0.516 / 0.535 |
| CP-DME | 0.495 / 0.411 | 0.370 / 0.293 | 0.814 / 0.942 | 0.516 / 0.531 |
| Bootstrap | 0.686 / 0.520 | 0.495 / 0.343 | 0.698 / 0.837 | 0.583 / 0.548 |
| CP-Bootstrap | 0.624 / 0.464 | 0.432 / 0.314 | 0.733 / 0.888 | 0.561 / 0.539 |

