# OpenReview forum: "Uncertainty-Calibrated Prediction of Randomly-Timed Biomarker Trajectories with Conformal Bands"
_NeurIPS.cc/2025/Conference — NeurIPS 2025 poster_

### Official Review · Reviewer_uuzK · 2025-06-29

**Clarity:** 4
**Significance:** 2
**Originality:** 2
**Rating:** 4
**Confidence:** 4

**Summary:**

The paper presents a nice application of conformal prediction, for the prediction of biotracker trajectories with random observation times.
The task is as follows: in an initial consultation (at time step t_0 = 0) we collect covariates $X$, potentially with some biomarker reading $Y_0$.
We then want to predict 'confidence bands' $C_t(X)$ for the trajectory $Y_t$ (for $t \in [0, T]$ or some discretization thereof) such that, given some random times $\mathcal{T} = \{T_1, ..., T_k\}$, it holds that
$\mathbb{P}[\forall t \in \mathcal{T},\ Y_t \in C_t(X)] \geq 1 - \alpha$ for some $\alpha$ chosen a priori.
The authors approach this via split conformal prediction techniques -- namely (vanilla) split conformal and group-conditional split conformal prediction -- with a conformity score designed for their setting.
They then go on to show the applicability of their approach in a real-world setting involving the prediction of brain biomarkers, highlighting the increased reliability when using their method, as well as the advantage of doing group-conditional CP with well-chosen covariates.
They conclude by showing how their uncertainty quantification can be used to confidently identify high-risk individuals, by building upon a metric that is reasonably well-established in the existing practice.

**Questions:**

My main concern is as stated above, and led me to give a borderline accept rather than a straight accept.
Could the authors perhaps elaborate more on either (i) how their methodology differs from existing CP methods besides the conformity score, and/or (ii) why the problem being solved would be of particular interest to the NeurIPS community, in contrast to the many other problems straight-forwardly solvable by CP? Are there any additional properties enjoyed by CP in the specific setting being considered?

**Ethical Concerns:**

["NO or VERY MINOR ethics concerns only"]

**Final Justification:**

One one hand, I think this is a nice applied paper; I particularly appreciate that rather than just doing conformal, the authors care about what benefits conformal guarantees actually give in practice. On the other hand, the method introduced has relatively little novelty, as noted during the discussion period. While the solid application makes me lean towards acceptance, this limited methodological novelty keeps me from giving the paper a score of 5.

**Limitations:**

Limitations were appropriately discussed.

**Paper Formatting Concerns:**

Please move the footnote mark in line 164 to somewhere it cannot be confused for math notation. I suggest adding it right after 'smallest value'.

**Quality:**

3

**Strengths And Weaknesses:**

The application presented in the paper is very nice, and the writing is easy to follow.
The methodology is sound, well-motivated and achieves good results.
My understanding of the specific example of brain biomarkers is limited, but the larger problem of biomarker prediction with random observation times is broadly applicable and relevant.

My main concern is that, at the end of the day, the proposed method reduces to simple split conformal and group-conditional split conformal prediction, treating the biomarkers at the random times as the labels. The only significant difference is a novel conformity score (which is a straightforward modification from existing ones, involving only adding a maximum over the time).

That said, the methodology and experimental validation are quite sound.
The section on group-conditional conformal prediction also cleanly presents the advantage of conditional guarantees. It's worth mentioning that the covariates used for the groups are some of the more comprehensive I have seen in applied conformal prediction (i.e., more than the usual applied CP paper), and clearly were carefully thought through.

I think it's also worth noting that there is some more recent work on CP that might be of interest to the authors. For example, stronger conditional guarantees than simple group-conditional CP can be attained via the work of https://arxiv.org/abs/2305.12616.
Also, similar ideas of involving maxima/suprema in the computation of the conformity scores have been considered in other contexts, e.g. in strategic CP (https://arxiv.org/abs/2411.01596) and in CP with adversarial robustness (e.g. https://arxiv.org/abs/2503.05239).

Finally, some minor nitpicks and suggestions:
- line 153-154: $\sigma$ might not be a function of (just) $\hat{Y}_t^{(i)}$, right?
- It would be best to make it clear in Problem 3.1 (line ~135) that though the guarantee cares only for $t \in \mathcal{T}$, the conformal bands are defined for all $t \in [0, T]$ (or some discretization thereof).
- Regarding the checklist:
    - Question 2 is marked as [TODO], but was answered with [YES] (judging by the justification).
    - Data&code: while I don't think it is essential that the code for this paper be reviewed, I strongly recommend that it be indeed published with the paper.
    - CIs on metrics: the authors claim in the checklist to provide CIs of all metrics, however I cannot find them for the tables with Precision, Recall & F1 (e.g. Table 1, and in the supplementary material). These can be easily computed e.g. with bootstrap, right?
    - Declaration of LLM usage is missing?

---

> ### Author Rebuttal · Authors · 2025-07-31
>
> We thank the reviewer for their extensive and constructive feedback. We are very glad that the reviewer found the application to be **well-motivated**, the methodology and experimental validation **sound**, and the writing **clear and accessible**. We also appreciate your positive comments on the **broad relevance** of predicting biomarkers under randomly timed observations, as well as our **carefully designed group-conditional covariates**, which you noted as **among the most comprehensive seen in applied conformal prediction** and go beyond what is typically explored in the literature.
>
> In summary, our response addresses the following **key points**:
> 1. We address all the suggestions (include the CIs in Table 1 and Table 4 of the paper)
> 2. We elaborate on the novelty of our work as well as the impact on the NeuRIPS community.
>
> ---
>
>
> **Weaknesses**
>
> - We agree with the reviewer that the normalizing function $\sigma$ does not depend only on $\hat{Y_t}^{(i)}$. A more complete notation would be $\sigma(\hat{Y_t}^{i},X_t;f)$, with $f$ denoting the predictor and $X_t$ the input at time $t$. This emphasizes that the uncertainty estimate depends not only on the predicted value $\hat{Y_t}^{(i)}$ but also on the covariates as well as properties of the predictor $f$ used to generate it. In the manuscript, we used the simplified notation $\sigma(\hat{Y}_{t}^{(i)})$ for readability. We will clarify in the final version that this is a simplified form and that the normalizing function is indeed dependent on other quantities.
>
>   To make this dependence concrete:
>     - For Gaussian Process-based predictors (e.g., DKGP, DME), $\sigma$  returns the posterior predictive standard deviation from the model.
>     - For frequentist ensemble methods (e.g., Bootstrap, Monte Carlo Dropout), $\sigma$ corresponds to the empirical standard deviation  across multiple forward passes or resampled models at time $t$ .
>
>
> - Thank you for the helpful comment. We will make the clarification in Problem 3.1 as you suggested.
> - Regarding your question about the CIs, in the revised manuscript  we have included the CIs in both tables. Here we present for reference the updated results for the Table 1 of the main manuscript:
>
> *Table: Youden-optimized discrimination on $z$-standardized rate of change (RoC) and lower-bound rate of change (LRoC) for MCI converters, with 95% bootstrap confidence intervals.*
>
> | Method       | Metric | $\tau^*$   | Precision        | Recall           | F$_1$             |
> |--------------|--------|------------|------------------|------------------|-------------------|
> | DRMC         | RoC    | $-0.006$   | **0.436 ± 0.022** | 0.671 ± 0.058    | 0.528 ± 0.023     |
> |              | LRoC   | $-0.012$   | 0.403 ± 0.022     | **0.884 ± 0.058** | **0.553 ± 0.023** |
> | CP--DRMC     | RoC    | $-0.006$   | **0.432 ± 0.022** | 0.740 ± 0.095    | 0.546 ± 0.024     |
> |              | LRoC   | $-0.020$   | 0.395 ± 0.022     | **0.915 ± 0.095** | **0.552 ± 0.024** |
>
>    We have also updated the Table 4 of Appendix accordingly.
>
> - Also we plan to release the code upon acceptance and we have made the modifications in the checklist.
>
> **Questions**
>
> We appreciate the opportunity to clarify the broader significance of our work. To the best of our knowledge, this is the **first conformal prediction framework tailored to randomly timed biomarker trajectories**, a setting that arises in clinical studies. Existing CP methods assume fixed or synchronized timepoints across subjects, which makes them incompatible with the asynchronous and irregular observation patterns present in real-world clinical data.
> While our nonconformity score is a key adaptation, our contribution goes beyond that: we formally extend split conformal prediction to the setting where time index \(\mathcal{T}\) is random. This includes a **normalization mechanism** that enables inference at arbitrary, unseen timepoints and supports guarantees even under irregular follow-up—a critical requirement for clinical deployment **with real data from clinical studies**. In Appendix F, we explain in detail how **methods designed for fixed-time trajectories cannot be applied in our setting**, which highlights the **importance of our results mainly from a practical (rather than a technical) perspective**.
> This problem setting is of particular interest to the **NeurIPS AI for Health and Medicine community**. Our approach not only enables **distribution-free guarantees** in this setting but also highlights real-world challenges (e.g., misaligned visit times, subgroup disparities, calibration under irregular sampling) that are underexplored in the CP literature.
> Additionally, our paper outlines several **methodological and applied directions** for future research—such as multivariate conformal prediction, adaptive methods for evolving trajectories, and group-conditional guarantees for intersectional subgroups—which we believe will help **bridge the gap between theory and practice** in clinical ML applications.

---

> > ### Comment · Reviewer_uuzK · 2025-08-03
> >
> > Thank you for your response. Unfortunately, I still have some concerns regarding the novelty.
> >
> > While the problem being tackled is of clear practical relevance -- as the authors mention, is immediately applicable to many clinical settings -- the method being proposed does seem to be standard split conformal prediction with a different conformity score.
> > To be more clear, as far as I can tell it is standard split conformal where the labels are given by $(Y\_t)\_{t \\in \\mathcal{T}}$.
> > Then the main novelty is to use the conformity score defined in Equation (2) of the paper.
> >
> > I do not understand, however, why the authors claim that "existing CP methods assume fixed or synchronized timepoints across subjects". I do not see how e.g. standard CP fails here. My best guess is that what they mean is that by using their conformity score the conformal bands can be tightened, which is probably true; however, the choice of conformity score used is a fairly immediate one.
> > It is possible that I am missing something, in which case I encourage the authors to clarify it.
> >
> > Given that the method itself reduces to standard CP, and that it is generally much easier to identify a setting in which standard CP works than one where it does not, I keep my score.

---

> ### Author Response · Authors · 2025-08-05
> **Response**
>
> Sorry for the misunderstanding. While it is true that our approach builds on the general split conformal prediction framework, our novel nonconformity score is essential for obtaining conformal bands for randomly-timed trajectories. In the following, we clarify why **standard CP methods cannot be applied to obtain valid *simultaneous* coverage guarantees at multiple random time points**, providing a more detailed explanation of Appendix F from the manuscript.
>
> The **standard approach** for obtaining simultaneous coverage over multiple time points would be to: i) apply split conformal prediction for each time point for which observations are available in the given dataset, and ii) **employ a union bound argument** to obtain trajectory-level coverage guarantees. However, **this standard strategy cannot be applied in our setting**, where each subject has a different number of observations occurring at different time points. More specifically,  there is no common set of time points across subjects over which a union bound could be applied. To adapt this naive approach to our setting, one could select a subset of the data corresponding to a fixed set of time points (e.g., 3, 7, 9, and 12 months) and extract only the corresponding observations from each subject. However, this leads to **substantial dataset reduction**. In a simple experiment following this approach, we found that only 37 trajectories with observations at all four selected time points remained out of the total dataset of 2200 trajectories. Beyond dataset reduction, this method **does not allow obtaining valid conformal bands for any test subject**. In particular, **since this standard approach provides conformal intervals only at the pre-specified time points $t$ (e.g., 3, 7, 9, 12 months), it cannot be used to provide valid confidence intervals for, e.g., a test subject who visits the clinic at months 4, 5, and 14**. Our normalized nonconformity score resolves that issue by incorporating the errors for all random time points into a single nonconformity score, which can be directly used to **design conformal bands for any test subject**. Thus, we provide conformal bands **with trajectory-level coverage guarantees without the need for union bounding**—which is not applicable to our setting as explained above.
>
> Thus, our nonconformity score::
> - Enables valid inference at **unseen and asynchronously sampled time points**;
> - Provides **trajectory-level coverage** without the need for a union bound.

---

> > ### Comment · Reviewer_uuzK · 2025-08-08
> >
> > I do not understand why the authors bring up an union bound. As I said in my review, just using vanilla CP works, no? No union bound here!
> >
> > Vanilla CP:
> > - Data $(X\_i, Y\_i)\_{i=1}^n$
> > - Choose conformity score $s : X \times Y \to \mathbb{R}$
> > - Compute scores on data: $S\_i = s(X\_i, Y\_i)$
> > - Compute adjusted quantile of scores: $t = \hat{q}_{\lceil (n+1) (1-\alpha) \rceil}(S\_1, \ldots, S\_n)$
> > - To predict: given a new $X$, produce $\\{y : s(X, y) \leq t\\}$.
> >
> > Proposed approach:
> > - Data $(X\_i, (Y\_{i,t})\_{t \in \mathcal{T}})\_{i=1}^n$
> > - Use proposed conformity score $s(X, (Y\_{t})\_{t \in \mathcal{T}}) = \max_{t \in \mathcal{T}} \frac{\lvert Y\_t - \widehat{Y}\_t \rvert}{\sigma(\widehat{Y}\_t)}$
> > - Compute scores on data: $S\_i = s(X_i, (Y\_{i,t})\_{t \in \mathcal{T}})$
> > - Compute adjusted quantile of scores: $t = \hat{q}_{\lceil (n+1) (1-\alpha) \rceil}(S\_1, \ldots, S\_n)$
> > - To predict: given a new $X$, produce $\\{(y\_{t})\_{t \in [0, T]} : s(X, (y\_{t})\_{t \in [0, T]}) \leq t\\} \cong (t \mapsto \widehat{Y\_t} \pm t \sigma(\widehat{Y\_t}))$.
> >
> > If anything, the only difference I could see (besides the choice of conformity score, which _is_ of some value, even if straightforward) would be in the definition of the predictive set, where in the proposed approach one would like to construct a function from all $t$, not only those in $\mathcal{T}$. But this seems like a minor detail; in fact, it has no impact on the proof. \
> > Please let me know if I am missing something.
> >
> > (Again, my position on the paper is mainly positive, just borderline due to being essentially just an instance of vanilla CP.)

---

### Official Review · Reviewer_FjGd · 2025-07-01

**Clarity:** 3
**Significance:** 2
**Originality:** 3
**Rating:** 5
**Confidence:** 3

**Summary:**

This paper introduces a method to estimate continuous biomarker trajectories with uncertainty bands from sparsely and irregularly sampled data, such as patient visits during a chronic disease journey.  The technical novelty is to use conformal methods, which previously rely on fixed time points, but to extend the formulation to accommodate irregular time points.  The authors demonstrate the method on an Alzheimer’s data set using standard image-derive biomarkers, hippocampal and ventricular volume.  Results show that a method that uses the lower bounds of the uncertainty bands more reliably detects progressors from MCI to AD than a standard method.

**Questions:**

What advantages does the conformal method offer over alternatives that can estimate uncertainty bands such as the Gaussian process model mentioned above?

**Ethical Concerns:**

["NO or VERY MINOR ethics concerns only"]

**Final Justification:**

It's early stage.  The extension to multiple biomarkers is essential for this to be relevant.  Still feel the paper lacks a bit the wider context of work on disease progression modelling - as mentioned in my original review - but I do think this paper is acceptable so increasing score one notch.

**Limitations:**

Fine

**Quality:**

3

**Strengths And Weaknesses:**

Strengths

The formulation seems nice, although I am not particular familiar with conformal methods.

The problem is real – both the reconstruction of biomarker trajectories from irregular time points and the estimation of uncertainty bands – and has genuine utility in clinical and application and not doubt other application areas.

Results show some potential benefit of the approach.

Weaknesses

There’s no comparison against other methods that can estimate uncertainty bands.  The obvious choice would be Gaussian process models.

Further to the point above, one limitation of the proposed method is that it only considers biomarkers individually.  Other work on disease progression modelling – see https://pubmed.ncbi.nlm.nih.gov/38191721/ for an overview – typically creates models of multiple biomarker trajectories on the same time axis, which is more powerful.  In particular, Lorenzi et al’s Gaussian Process disease progression model https://www.sciencedirect.com/science/article/abs/pii/S1053811917307061, considers multiple biomarkers with uncertainty bands so seems on the face it more advanced than the proposed method.

---

> ### Author Rebuttal · Authors · 2025-07-31
>
> We thank the reviewer for their feedback and their thoughtful review. We are happy that the reviewer found our problem—reconstructing biomarker trajectories from **irregular time points and quantifying uncertainty** through prediction intervals—**genuinely useful for clinical and other application areas**, and saw the **potential benefit** of our approach.
>
> In summary, our response addresses the following **key points**:
> 1. We explain how our conformal method **compares with five baseline uncertainty quantification methods** (e.g., Gaussian-process based bands) based on our comparative case from Section 4.
> 2. We elaborate on how our conformal algorithm can be **extended to account for multiple biomarkers**.
>
> ---
>
> **Weaknesses**
>
> - Our current comparison in Section 4 consists of a total of five baselines, including Gaussian Process-based baselines—see also Appendix for comparative experiments for Sections 5 and 6. Specifically, both **DKGP** (Deep Kernel Gaussian Process) and **DME** (Deep Mixed Effects model) are GP-based methods capable of estimating uncertainty bands. In figure 2 you can notice that we compare the DKGP and DME with their conformalized counterparts, where our approach is able to achieve the nominal coverage with tighter bands.
>
> - While our current implementation focuses on single-biomarker prediction bands, our conformal algorithm can be directly extended to multivariate predictors that jointly model multiple biomarker trajectories. As noted in the future work section, we are actively pursuing this direction. A natural generalization involves defining the nonconformity score as the maximum normalized residual across all biomarkers and timepoints, allowing for multivariate trajectory-wise conformal coverage. More specifically, this means that in the current expression of the nonconformity scores we would add an additional index $j=1,...,B$, ranging over all $B$ biomarkers we care about and the same index would be added in the true and predicted biomarker values, which will now be denoted by $Y_{t,j}^{(i)}$ and $\hat{Y}_{t,j}^{(i)}$, respectively.
>
> Note also that our method is not intended to substitute specific predictive models, but rather to provide a conformalization procedure that can be applied to **any base predictor**—including multivariate models such as the Gaussian Process disease progression framework proposed by Lorenzi et al. While GP-based models are powerful, they come with limiting assumptions (e.g., Gaussianity, stationarity) that may not hold in clinical data. Our conformal approach complements these models by providing **distribution-free coverage guarantees** in the presence of model misspecification.
>
> **Questions**
>
> GP-based methods estimate uncertainty based on modeling assumptions, but often produce overly conservative intervals, as shown in Figure 2 for DKGP and DME. Frequentist methods like Bootstrap and Monte Carlo Dropout can underestimate uncertainty and lead to narrow, miscalibrated bands. Our conformal method improves both cases: it can be applied on top of any of these predictors to produce **narrower or better-calibrated intervals** that achieve valid coverage guarantees. Unlike model-based or sampling-based methods, our conformal prediction method makes no distributional assumptions and provides formal coverage guarantees that hold by construction.

---

> > ### Comment · Reviewer_FjGd · 2025-08-05
> > **Brief response**
> >
> > Thanks for the rebuttal.  Acknowledged that section 4 does include a GP model.  I was thinking of section 6, but I guess it does make the point that conformalization generally improves the band estimates.
> >
> > My main concern remains about putting this work in the wider context of disease progression modelling, but I do believe this work has potential utility in that area.

---

> > > ### Author Response · Authors · 2025-08-05
> > > **Response**
> > >
> > > We thank the reviewer for their follow-up and appreciate the opportunity to further clarify this point. In Section 6 of the main paper, we focus on the DRMC predictor for illustrative purposes. However, for completeness, we provide results from all evaluated predictors—including DKGP and DME, both of which are Gaussian process-based—in Table 4 of Appendix I.
> > >
> > > We would also like to emphasize that our conformal algorithm is model-agnostic and thus can be applied on top of any base predictor, including disease progression models such as those cited by the reviewer, provided that the necessary assumptions hold. Extending our conformal approach to multivariate disease progression models is a promising direction, and we are currently exploring this as part of our future work.

---

### Official Review · Reviewer_Ragn · 2025-07-01

**Clarity:** 3
**Significance:** 3
**Originality:** 3
**Rating:** 5
**Confidence:** 4

**Summary:**

This work proposes a conformal prediction method for longitudinal biomarker trajectories with irregular follow-up times. The proposed time-adaptive nonconformity scores leverage uncertainty estimates to enable conformal predictions of trajectory values at arbitrary time points. The method is demonstrated to outperform baseline approaches in terms of coverage and interval width on brain biomarker trajectories.
To address population heterogeneity (e.g., gender, race, and existing risk factors), the authors propose stratifying the calibration data by relevant groups. This approach is demonstrated to attain nominal coverage, unlike their unconditional (non-stratified)  counterparts.
To further demonstrate the clinical utility, the authors propose a *lower-bound rate of change* (LRoC) derived from the lower bound of the conformal prediction band, as a tool for identifying patients at high risk of Alzheimer’s disease.

**Questions:**

1. Figure 2, what the mean interval width is averaging over, the subject index $i$, the time index $t$, or both? If averaging over $t$ is involved, does the irregularly sampled $t$ affect the result?

2. Figure 4, while coverage is attained, why is it further away for the nominal level compared to those in figure 2? Is this due to the baseline prediction model $\hat Y$ and $\sigma$ not using the grouping $G$? If so, why the baseline models not using them?

3. Line 349, how is the prediction horizon $t_N$ specified? Further, RoC's current definition in line 361 -- 364 is ambiguous. It is beneficial to add formula for RoC as well, preferably side-by-side to that of LRoC in (3).

4. Line 370, trial eligibility? Isn't this $\tau$ is just used for identifying high-risk or not?

5. While Table 4 in the appendix includes all results utilizing different base predictors, why Table 1 only shows DRMC?

6. Is it possible to also show threshold $\tau^*$ in certain way in figure 5 for better illustration?

**Ethical Concerns:**

["NO or VERY MINOR ethics concerns only"]

**Final Justification:**

The authors addressed my initial concern regarding the registration problem during the rebuttal. While acknowledging that this issue lies beyond the scope of the current manuscript, they agreed that future investigation in this area could be valuable. Additionally, the authors provided supplementary results employing threshold-free evaluation to further demonstrate the advantages of the proposed LRoC method. They also clarified several implementation details and notations in response to my initial questions. I thank the authors for their prompt and thorough replies. Overall, my assessment of this manuscript is positive.

**Limitations:**

The manuscript lacks a dedicated discussion of limitations, despite Section 8 providing some directions for future work. While these future directions may implicitly highlight constraints of the current approach, a more explicit and constructive elaboration of limitations, particularly in light of the discussed weaknesses, would be beneficial.

**Paper Formatting Concerns:**

Current responses to checklist item 2., 8., and 16. is inadequate. Authors need to address them more carefully. This is a minor point, though.

**Quality:**

3

**Strengths And Weaknesses:**

**Strength:**

1. This work presents a novel extension of the conformal prediction framework to handle longitudinal data with irregularly spaced time points, a critical advantage in clinical and biomedical applications where sampling times are often inconsistent across patients. The reviewer agrees that this addresses an important limitation of existing approaches, which typically require uniformly sampled time points for all trajectories.

2. The proposed method achieves nominal coverage with tight prediction bands compared to baseline approaches that rely on model-specific uncertainty estimates. This is demonstrated on a brain biomarker example, where the necessity of uncertainty calibration—particularly in the presence of population heterogeneity—is clearly illustrated through comparisons with baseline and/or non-stratified methods.

**Weaknesses:**

1. The method relies on time-wise uncertainty estimates to construct normalize nonconformity scores, yet it does not explicitly account for registration problem (e.g., misalignment due to irregular or asynchronous sampling schedules). Such misalignment could distort both the estimated mean trajectory and the uncertainty estimates, which assume a consistent temporal structure. It is unclear whether the authors considered how registration (or lack thereof) might affect the validity of these uncertainty estimates in their data analysis. This omission raises questions about whether the current approach adequately captures the true variability in trajectory dynamics or whether the results could be sensitive to unaccounted temporal misalignment. Future work could explore the impact of registration problem on the proposed uncertainty-aware predictions.

2. The specification of groups (e.g., line 263) is not thoroughly discussed in the manuscript, despite being a critical modeling choice that could influence results. See also my later Question 2. The authors could benefit from a deeper discussion in Section 8.

3. The contrast between RoC (rate of change) and LRoC could have been more effectively leveraged to highlight the clinical value of the proposed uncertainty-aware metrics. However, the current presentation (e.g., Table 1) may not fully clarify such advantage, as the gain in recall could stem from either reduced precision or improved discriminative power. Alternative summaries, such as ROC-AUC or PR-AUC, would provide a more comprehensive evaluation of performance across thresholds, rather than focusing solely on a single Youden-optimized $\tau^*$ point.

---

> ### Author Rebuttal · Authors · 2025-07-31
>
> We thank the reviewer for their feedback and for highlighting the key strengths of our work. We are delighted that reviewer recognizes that we **address a critical limitation of prior methods**, while achieving nominal coverage with tight prediction bands compared to baseline approaches. We are also grateful for the reviewer’s appreciation of our empirical demonstration on brain biomarkers and the importance of uncertainty calibration in heterogeneous clinical populations.
>
> In summary, our response addresses the following **key points**:
> 1. We explain that our conformal method is designed so that it can directly handle irregular trajectories without the need of temporal alignment.
> 2. We expand and enhance the comparison between the RoC and LRoC with the requested metrics.
>
> ---
>
> **Weaknesses**
>
> - In our setting, each subject's time axis is defined relative to their baseline visit, and our predictors are trained to forecast future biomarker values based on this subject-specific temporal reference frame. Also, we would like to clarify that our method is specifically designed to handle trajectories that are not temporally aligned across subjects. In our formulation, the set $\(\mathcal{T}\)$ of time points at which observations are collected for a given subject is treated as a **random variable**, and our coverage guarantee in **Theorem 3.2** is stated with respect to the joint distribution over the triplet $\((X, \mathcal{T}, Y)\)$. This formulation explicitly incorporates the randomness of observation times **without relying on any assumption of a shared or aligned time axis across subjects**. Furthermore, our nonconformity scores are computed at subject-specific timepoints relative to each individual's baseline, and we do not perform any form of temporal registration or warping. The validity of our coverage guarantee holds under this irregular and asynchronous observation setting and is rigorously proven in **Appendix B**.
> - We thank the reviewer for raising this point regarding group specification. To clarify, the grouping function $\( G: \mathbb{R}^d \rightarrow \mathcal{G} \)$, introduced in line 263, is instantiated using single categorical covariates in our current experiments. As described in Section 5.2, we evaluate group-conditional conformal prediction across five clinically relevant stratifications: sex, race, education level, diagnosis, and APOE4 allele status. These covariates are well associated with Alzheimer’s Disease and Brain Ageing  [1,2]. We will elaborate about this in the revised manuscript.
> -  We thank the reviewer for the insightful suggestion to contrast RoC and LRoC using threshold-free evaluation metrics. In response, we conducted a comprehensive analysis across ten trajectory models (e.g., DKGP, CP-DRMC, Bootstrap), using **partial AUC** (PrAUC) computed at varying sensitivity thresholds. Rather than relying on full ROC-AUC, which can obscure performance at clinically relevant operating points, we report **partial AUC at 0.8 sensitivity**, a threshold that offers a strong balance between recall and precision. A sensitivity of 0.8 is clinically appropriate for high-risk screening applications such as early MCI-to-AD conversion detection—prioritizing early identification of converters without overwhelming clinicians with false positives. The difference in partial AUC between RoC and LRoC is negligible for DKGP (Δ = −0.003), reverses in favor of LRoC for DQR (Δ = +0.007), and even improves substantially for DRMC (Δ = +0.032). This supports our claim that **LRoC can enhance conservative prediction under uncertainty without meaningfully degrading AUC-based performance**. These findings clarify that LRoC does not aim to outperform RoC across all metrics, but rather serves a complementary role—offering robust sensitivity under worst-case assumptions. We have included these comparisons in Appendix Table 4 and updated the manuscript to emphasize the 0.8-sensitivity PrAUC as a clinically justified and empirically stable operating point. We will include the additional metrics in Table 4 of Appendix I.
>
> **Questions**
>
> - The mean interval width reported in Figure 2 is computed over both indices $i$ and $t$, as briefly mentioned in line 227 of the manuscript. The averaging over $t$ is necessary for a fair empirical evaluation of our method, since the probability distribution in our coverage guarantee is over triplets (X,$\mathcal{T}$,Y), where $\mathcal{T}$ is the set of random points corresponding to a test subject. Our method is specifically designed to handle irregularly sampled trajectories, and the mean interval width we compute provides the empirical average over random subjects with different sets of random time points.
>
> - The coverage differences observed in Figure 4, relative to Figure 2, are primarily due to the use of a **common calibration set** across the population and group-conditional conformal prediction experiments. Specifically, we used a fixed calibration set comprising 20% of the subjects, which was selected based on the calibration set size exploration for the Population CP (see Appendix G.2, and applied the same set when evaluating Group-Conditional CP. This decision was made to ensure a **fair comparison** between population and group-conditional methods under consistent data conditions. However, we would not expect the calibration set size that we selected for the Population CP to also provide tight intervals for the Group-Conditional CP. **In practice, if we were to apply Group-Conditional CP, we would make another exploration (similar to that in Appendix G.2) for the calibration set size that would result in tight bands for the individual groups.**   In particular, we would select the calibration set size **separately for each group** (e.g., CN, MCI, AD) to achieve tighter adherence to the nominal level.
> - The prediction horizon is defined individually for each subject as the time span between their baseline and last available observation. On average, this corresponds to **43 months** (standard deviation: **32 months**), but the horizon varies across subjects due to irregular follow-up schedules. We will clarify this point in the revised manuscript and will also **update the notation** throughout the paper to make it explicit that both $t_0$ and $t_N$ are **subject-specific** (i.e., indexed by subject $i$).
>
>     We agree that the mathematical definition of Rate of Change (RoC) in lines 361–364 should be included in the manuscript. In the revision, we will provide an explicit formula for RoC and place it side-by-side with the LRoC definition in Equation (3). Specifically, we define RoC for subject $i$ as:
> $$
> \operatorname{RoC}_i = \frac{\hat{y}_i(t_N^{(i)}) - y_i(t_0^{(i)})}{t_N^{(i)} - t_0^{(i)}}
> $$
> Where:
> - $t_0^{(i)}$ and $t_N^{(i)}$ denote the subject-specific baseline and final timepoints,
> - $\hat{y}_i(t)$ is the predicted biomarker value at time $t$.
>
> - In our framework, the term “trial eligibility” refers to identifying high-risk subjects likely to experience rapid decline toward Alzheimer’s disease. Such subjects are eligible to be included in a clinical trial and therefore the terms “trial-eligible” and “high-risk” subjects are used interchangeably in the manuscript. In the revised manuscript, we will make sure to clarify the above.
>
> - Table 1 focuses on DRMC as a representative baseline due to space constraints in the main manuscript. However, as shown in Appendix I Table 4, the same conclusion holds across all base predictors—namely, that LRoC improves recall relative to RoC, often with comparable or improved F1 scores. We will include the full set of results in the main paper in the final version where space permits it.
>
> - We will update the corresponding plot to include a decision boundary line with slope equal to the Youden-optimized threshold. This line will clarify how the threshold corresponds to the separation criterion in the LRoC and CP-LRoC.
>
> **Limitations**
>
> Indeed, we do not explicitly discuss limitations of the current approach (only our section on future work hints to open problems). We will include a dedicated paragraph on limitations in the final version of the paper.
> One key limitation of our current group-conditional approach is its **lack of scalability** with respect to the number of covariates. While stratifying by a single covariate (e.g., sex, race, diagnosis) allows for interpretable subgroup analysis, it becomes impractical to condition on **combinations of multiple covariates** as this may lead to very small subgroup sample sizes (e.g., Asian, Homogenous for APOE4). Even though our method can be applied to such subgroups, in practice we expect that small sample sizes corresponding to underrepresented subgroups will yield uninformative conformal bands. This restricts the practical applicability of the analysis and may overlook intersectional disparities in model calibration. As discussed in our future work, we aim to extend the group-conditional procedure to **intersectional subgroups** (e.g., Black females aged 55–65), which would provide more clinically relevant and equitable coverage guarantees. Another limitation is that our current framework cannot be directly applied to external datasets while preserving the original coverage guarantees, unless a new calibration step is performed on the target study. When applied to external studies with potentially different demographic distributions, data acquisition protocols, or diagnostic criteria, the assumptions for valid CP are violated. Therefore, without prior recalibration, applying our method to unseen cohorts may lead to invalid or misleading prediction intervals.
>
> [1] Corder, E. H., et al. (1993). Gene dose of apolipoprotein E type 4 allele and the risk of Alzheimer's disease in late onset families
>
> [2] Meng, X., et al. (2012). Education and dementia in the context of the cognitive reserve hypothesis: a systematic review with meta-analyses and qualitative analyses

---

> > ### Comment · Reviewer_Ragn · 2025-08-05
> >
> > Thank you for the detailed, point-by-point response. I appreciate the effort invested in addressing the concerns and for providing additional results within such a short timeframe.
> >
> > I'd like to slightly clarify my previous comments on registration.
> > I acknowledge that the proposed method does not assume a common sampling grid and that the nominal coverage is achieved. The earlier comment concerning registration were primarily intended to highlight that phase misalignment can also arise from data-related factors. For example, the baseline time for two patients may correspond to different stages of disease progression, leading to temporal misalignment in their trajectories.
> > Such misalignment, if not accounted for, has the potential to affect both predictions and uncertainty estimates (i.e., $\hat Y$ and $\sigma$). From this perspective, the reviewer was curious whether incorporating a registration step during data pre-processing could further improve performance, perhaps leading to even tighter uncertainty bands. That said, it is also possible that such issues were already addressed by the cohort's design, particularly in the definition of inclusion time.
> >
> > Though, the reviewer is not requesting additional results or experiments at this point. The authors are, of course, welcome to provide any further comments if they wish.

---

> > > ### Author Response · Authors · 2025-08-05
> > > **Response**
> > >
> > > We sincerely thank the reviewer for the thoughtful clarification regarding registration and the potential impact of disease stage misalignment. We appreciate the distinction drawn between irregular sampling and the possibility that baseline visits may correspond to different stages of disease progression across individuals.
> > >
> > > Disease stage misalignment is not explicitly addressed in our current conformal framework, as this lies outside the scope of our present work. While our predictors are conditioned on subject-specific covariates, we do not currently incorporate alignment based on latent disease severity.
> > >
> > > The reviewer raises an important point regarding the potential benefit of incorporating a registration step during data pre-processing—specifically, aligning subjects by underlying disease stage to reduce temporal misalignment. We note that disease stage is distinct from clinical diagnosis, and unlike diagnosis, it is inherently latent and not directly observable. Translating conformal uncertainty from the biomarker space to the disease stage space is therefore a nontrivial challenge. Integrating latent disease staging into the conformal prediction framework—while preserving formal coverage guarantees—is an open and promising direction that we intend to pursue in future work.
> > >
> > > We will revise the manuscript to better reflect this important consideration in the discussion of limitations and future directions. We again thank the reviewer for their constructive feedback and engaged assessment throughout the review process.

---

> > > > ### Comment · Reviewer_Ragn · 2025-08-09
> > > >
> > > > Right, I now agree that conformal uncertainty from the biomarker space to the disease stage space would be a separate topic.
> > > >
> > > > Thank you for the prompt and thorough replies. Overall, my assessment of this manuscript is positive.

---

### Official Review · Reviewer_W7CN · 2025-07-02

**Clarity:** 3
**Significance:** 2
**Originality:** 2
**Rating:** 3
**Confidence:** 4

**Summary:**

This paper aims to construct conformal prediction bands for randomly timed trajectories, where outcomes are observed only at random time points and may neither be consecutive nor follow any fixed time grid. The authors propose to utilize normalized nonconformity scores originally introduced in [Yu et al.], in conjunction with the standard split conformal prediction pipeline. Furthermore, they extend their study to group-conditional conformal prediction, leveraging the method from [Romano et al.]. The proposed methodology is applied to two well-established longitudinal clinical datasets. Additionally, the authors introduce a new metric based on conformal prediction bounds to assess the risk of progression from Mild Cognitive Impairment (MCI) to Alzheimer’s disease.

[Yu et al.] Xinyi Yu, Yiqi Zhao, Xiang Yin, and Lars Lindemann. Signal temporal logic control synthesis among uncontrollable dynamic agents with conformal prediction.
[Romano et al.] Yaniv Romano, Rina Foygel Barber, Chiara Sabatti, and Emmanuel Candès. With malice toward none: Assessing uncertainty via equalized coverage

**Questions:**

1. Compared to [Yu et al.], what unique challenges arise in extending the method to randomly-timed observations, aside from substituting the predictive model with one capable of handling such data?
2. In Figure 2, could you please report the standard deviations or confidence intervals associated with the coverage estimates?

**Ethical Concerns:**

["NO or VERY MINOR ethics concerns only"]

**Final Justification:**

The authors addressed my concerns regarding novelty during the discussion period and provided helpful answers to my questions. Accordingly, I have raised my score to 3. That said, I still have some reservations, as in my view the contribution—while valuable—offers only a modest advance over the method proposed in [Yu et al.] in the context of the broader conformal prediction literature, which makes me hesitant to fully recommend acceptance at NeurIPS.

**Limitations:**

No, please refer to the weaknesses section for further details.

**Quality:**

2

**Strengths And Weaknesses:**

Strengths:
The paper is clearly written, well-structured, presents a clear problem statement, and is easy to follow.

Weaknesses:
1. Novelty: My main concern lies with the paper's technical novelty. The proposed method essentially applies existing conformal prediction frameworks (as introduced in [Yu et al.]) to the setting where outcomes are observed at irregular, random time points. The normalized nonconformity score presented in Equation (2) appears to be identical to that of [Yu et al.], and the overall methodology follows the standard split conformal prediction paradigm. Similarly, the group-conditional conformal procedure mirrors the approach in [Romano et al.] without apparent modification. That being said, while the setting of randomly-timed observations is practically relevant and underexplored, the methodological contribution appears to be more an adaptation rather than an innovation. If there are subtle but important theoretical or algorithmic contributions to handling the randomness in time stamps, I encourage the authors to clearly articulate and highlight them.
2. Empirical Evaluation: Some aspects of the experimental results are counterintuitive. If I have misunderstood, I would appreciate clarification. In particular, regarding Figure 4: when reporting stratified coverage by features such as Race, did you partition the test set into mutually exclusive and collectively exhaustive groups (e.g., Asian, Black, and White)? If so, it is puzzling that the conformal coverage appears to fall below the nominal level (e.g., 90%) for all subgroups. If each subgroup undercovers, then by the law of total probability, the overall marginal coverage would also be below target - contradicting the guarantees of marginal conformal prediction. A more detailed discussion or clarification of this point would be helpful.
3. Limitations: The paper does not provide a dedicated discussion on its limitations.

---

> ### Author Rebuttal · Authors · 2025-07-31
>
> We would like to thank the reviewer for acknowledging that our paper is **well structured** and **clearly written**.
> In summary, our response addresses the following **key points**:
> 1. **Major concern:** We elaborate on the contributions of our conformal method based on our detailed comparison in Appendix F. We explain that our method: i) is applicable to randomly-timed trajectories via the **novel use of a normalizing function**, ii) provides **provable coverage guarantees in the context of randomly-timed trajectories**, and iii) is very **impactful for practical applications with randomly-timed trajectories**, as approaches for fixed-time trajectories cannot be applied to such settings.
> 2. We provide updated results that  address the issue pointed out by the reviewer regarding performance disparities across groups in population conformal prediction.
> 3. As requested, we discuss the limitations of our approach, and we have already revised the manuscript accordingly.
>
> In our response below, we aim to address in detail each of these concerns, along with several additional questions related to the experimental results.
>
> ---
>
> **Weaknesses**
>
> - Our core technical contribution is the introduction of a **normalizing function** that enables conformal prediction for randomly timed trajectories. This adaptation allows nonconformity scores to be evaluated at arbitrary, subject-specific timepoints—supporting valid coverage even under irregular and asynchronous sampling.   The normalizing function enables us to compute a normalized prediction error across all time steps, ensuring that no component within the maximum dominates the others in terms of scale. The above are explained in detail in Appendix F, but in the revised manuscript we will provide intuition for the reader earlier in Section 3. Thanks to this methodological advance, our conformal method is the **first to provide formal coverage guarantees for the setting of randomly-timed trajectories**.
> Beyond its theoretical contributions, our approach addresses a critical practical gap: it is, to our knowledge, the first to implement conformal prediction in the setting of **randomly-timed trajectories**. Prior conformal methods (e.g., Stankeviciute 2021, Yu 2023, Cleaveland 2024) assume fixed, synchronized time grids across subjects, making them unsuitable for real-world clinical data where observation schedules vary widely. Our framework removes this constraint, unlocking conformal prediction for a broad range of applications in longitudinal healthcare modeling.
>
> - We thank the reviewer for pointing out the discrepancy in stratified coverage. Upon review, we identified that when we evaluated the population conformal prediction per race-group we had excluded several subjects due to fallacious filtering.
>  We sincerely thank the reviewer for catching this issue. We revised the figure accordingly and below we attach the results for the Race. The rest of the results are unaffected.
> We add the corrected results too as a table.
>
> *Table: Comparison of coverage under population vs. group-conditional conformal prediction, stratified by race (White, Black, Asian)*
>
> | Race   | Population Conformal Prediction (%) | Group-Conditional Conformal Prediction (%) |
> |--------|--------------------------------------|--------------------------------------------|
> | White  | 92.0                                 | 92.0                                       |
> | Black  | 88.0                                 | 92.0                                       |
> | Asian  | 86.0                                 | 96.0                                       |
>
> - Indeed, we do not explicitly discuss limitations of the current approach (only our section on future work hints to open problems). We will include a dedicated paragraph on limitations in the final version of the paper. One key limitation of our current group-conditional approach is its **lack of scalability** with respect to the number of covariates. While stratifying by a single covariate (e.g., sex, race, diagnosis) allows for interpretable subgroup analysis, it becomes impractical to condition on **combinations of multiple covariates** as this may lead to very small subgroup sample sizes (e.g., Asian, Homogenous for APOE4). Even though our method can be applied to such subgroups, in practice we expect that small sample sizes corresponding to underrepresented subgroups will yield uninformative conformal bands. This restricts the practical applicability of the analysis and may overlook intersectional disparities in model calibration. As discussed in our future work, we aim to extend the group-conditional procedure to **intersectional subgroups** (e.g., Black females aged 55–65), which would provide more clinically relevant and equitable coverage guarantees. Another limitation is that our current framework cannot be directly applied to external datasets while preserving the original coverage guarantees, unless a new calibration step is performed on the target study. When applied to external studies with potentially different demographic distributions, data acquisition protocols, or diagnostic criteria, the assumptions for valid CP are violated. Therefore, without prior recalibration, applying our method to unseen cohorts may lead to invalid or misleading prediction intervals.
>
> **Questions**
>
> -  We would like to clarify that our contribution goes beyond simply changing the predictive model. The key methodological novelty lies in the introduction of a **time-aware normalizing function** used in the nonconformity score, which enables conformal prediction in the context of **randomly timed longitudinal trajectories**. Existing methods, such as *Yu et al.* (2023), assume that all subjects share a common set of fixed observation times across the training, calibration, and test sets. These approaches define normalization terms `σ_t` only at pre-specified time points and therefore provide valid conformal coverage **only at those points**, limiting their applicability to real-world data where visits occur at variable and often irregular times. Attempting to apply such methods to real biomarker trajectories would necessitate discarding a large portion of the dataset — e.g., in our experiments, selecting four fixed time points (3, 7, 9, 12 months) yielded only 37 usable trajectories out of 2200.
> To address this, we introduce a **normalizing function `σ(·)`** that produces predictive uncertainty estimates `σ(Ŷ_t)` at **any time point `t`**, even if no observations exist at that time in the calibration set. This allows us to compute normalized nonconformity scores `|Ŷ_t - Y_t| / σ(Ŷ_t)` at **arbitrary, subject-specific test-time points**, thus supporting valid coverage under asynchronous and irregular sampling schedules. Crucially, this normalization ensures that no individual time point dominates the score due to scale differences, and that prediction intervals adapt to time-varying uncertainty. We explain the full technical details of this construction in **Appendix F**, but in the revised manuscript, we will provide **earlier intuition in Section 3** to help guide the reader through this key innovation.
>
>
>
> - The confidence intervals are also visualized in the Figure 2a. This is the table with the 95% CI of the coverage metric across the folds:
>
> *Table: 95% confidence intervals for marginal coverage and average interval width across methods, before and after conformalization.*
>
> | Method         | Coverage CI           | Interval Width CI      |
> |----------------|------------------------|-------------------------|
> | DKGP           | [0.9628, 0.9980]       | [1.8508, 1.8517]        |
> | DME            | [0.9070, 0.9687]       | [1.9554, 1.9568]        |
> | DQR            | [0.4099, 0.5380]       | [0.6132, 0.6510]        |
> | DRMC           | [0.4227, 0.5513]       | [0.6524, 0.7467]        |
> | Bootstrap      | [0.3675, 0.4073]       | [0.4771, 0.4915]        |
> | CP-DKGP        | [0.8758, 0.9486]       | [1.2311, 1.2317]        |
> | CP-DME         | [0.8567, 0.9346]       | [1.5976, 1.5987]        |
> | CP-DQR         | [0.8584, 0.9364]       | [1.3722, 1.4677]        |
> | CP-DRMC        | [0.8584, 0.9364]       | [1.6931, 1.9319]        |
> | CP-Bootstrap   | [0.8697, 0.9442]       | [1.9360, 2.1190]        |

---

> ### Comment · Reviewer_W7CN · 2025-08-06
>
> I thank the authors for their response, including corrected experimental results and the detailed explanation of the novel normalizing function $\sigma(\cdot)$ that produces predictive uncertainty estimates $\sigma(\hat{Y}t)$ at any time point $t$ for those time points that are missing. I acknowledge that this is a novelty that I overlooked before, and appreciate the authors' agreement on highlighting this key novelty in their main body for future readers. A quick follow-up question: Did you use $\mathcal{D}_{\text{train}}$ or $\mathcal{D}_{\text{cal}}$ to train the normalizing function $\sigma(\cdot)$?
>
> I’m tentatively considering increasing my score to borderline reject (or potentially higher) in light of the novelty of using a predictor function to estimate the normalizing score, but I would prefer to wait for the authors’ next round of responses before making a final decision. The reason I remain slightly negative is that, in my personal view, the significance of this addition to the broader conformal prediction literature is not sufficient for a recommendation to NeurIPS. Please feel free to correct me or disagree, my understanding of the approach proposed by [Yu et al.] is that it can be implemented even without the normalization term, i.e., let $S_i = \max_{t \in [T]} |\hat{Y}_t^{(i)} - Y_t^{(i)}|$, which corresponds to the maximum residual one could make across the observation window for trajectory $i$. This score (without normalization) preserves the coverage guarantee. The normalization term is more optional and is more for improving adaptivity and efficiency (width) of their resulting conformal prediction bands. Without normalization, the core intuition behind the nonconformity score design appears to be the same between [Yu et al.] and this paper, regardless of the randomness in observing/missing values at particular time steps.
>
> Minor: There is a missing citation on line 761 in Appendix F

---

> > ### Author Response · Authors · 2025-08-08
> > **Response**
> >
> > We thank the reviewer for their thoughtful comments. Below we address each point.
> >
> > All our experiments use predictive models equipped with inherent uncertainty functions, $\sigma(\cdot)$. For Gaussian process–based models such as DKGP and DME, $\sigma(\cdot)$ corresponds to the posterior predictive standard deviation obtained via exact inference. For other models, such as Deep Regression with Monte Carlo Dropout (DRMC) and Bootstrap ensembles, $\sigma(\cdot)$ is the sample standard deviation computed across multiple stochastic forward passes (DRMC) or multiple models (Bootstrap). This is described in Appendix E.2, but we will make it explicit in the main manuscript that $\sigma(\cdot)$ is estimated solely from the training set $D_{\text{train}}$. For models without an inherent uncertainty estimate, $\sigma(\cdot)$ would be learned from $D_{\text{train}}$; the calibration data cannot be used for this purpose, as it would violate the i.i.d. assumption for the nonconformity scores $R^{(i)}$.
> >
> > It is true that the approach proposed by Yu et al. 2023 can be implemented even without the normalization term. However, the time-adaptivity of our conformal intervals owing to the **normalization term is critical to obtaining informative conformal bands**. Without normalization, the interval width would be fixed across time and dictated by the largest prediction errors—typically from long-horizon forecasts, which are inherently more difficult. In our dataset, some trajectories span more than 140 months, so large long-term errors would propagate to short-term predictions, making them overconservative. Clinically, this is problematic: overconservative short-term bands inflate the Lower-bound Rate of Change (LRoC) metric, which we use to flag high-risk subjects. This would raise recall but substantially reduce precision by misclassifying many MCI-stable individuals as progressors. By scaling residuals with $\sigma(\cdot)$ at each time, our method avoids this behavior—intervals remain appropriately tight for short-term predictions while widening for long-term ones—thereby preserving both the validity of the coverage guarantee and the clinical utility of derived risk metrics such as LRoC.

---

> > > ### Comment · Reviewer_W7CN · 2025-08-09
> > >
> > > I thank the authors for their reply, which addressed my questions. I would therefore like to raise my score accordingly.

---

### Decision · Program_Chairs · 2025-09-17

**Decision:**

Accept (poster)

**Comment:**

This paper presents a method for constructing conformal prediction bands for longitudinal biomarker trajectories observed at irregular and random time points, addressing a common challenge in clinical data where visits do not follow a fixed schedule. The approach builds on normalized nonconformity scores introduced in Yu et al. and employs standard split conformal prediction, further extending to group-conditional conformal prediction following Romano et al. The methodology is demonstrated on Alzheimer’s disease biomarkers (e.g., hippocampal and ventricular volume) using two well-established clinical datasets, with a proposed lower-bound rate of change (LRoC) metric to identify patients at higher risk of progression from MCI to Alzheimer’s.

Strengths noted across reviews include the paper’s clarity, structure, sound methodology, and practical relevance for clinical applications. The approach is recognized for adapting conformal prediction to irregular time points, achieving nominal coverage with relatively tight prediction bands, and carefully considering covariates in group-conditional analysis. Reviewers highlight the method’s potential utility in reliably detecting disease progression, its applicability to heterogeneous populations, and its straightforward yet effective application of conformal principles. However, multiple weaknesses are also raised. Several reviewers question the level of technical novelty, noting that the method largely adapts existing frameworks (split conformal, group-conditional conformal, normalized scores from Yu et al.) with only minor modifications. Others flag omissions: the lack of discussion on trajectory registration/misalignment problems, the absence of comparison to established alternatives, and the restriction to single biomarkers rather than joint modeling across multiple trajectories. Empirical concerns are also raised, including seemingly inconsistent subgroup coverage results (potentially contradicting marginal guarantees), insufficient discussion of how groups were specified, and underdeveloped analysis of the added value of LRoC relative to other metrics. Minor issues include missing confidence intervals on reported metrics, absent code release, and missing declarations in the checklist. Overall, reviewers find the application relevant and promising but encourage stronger articulation of novelty, deeper evaluation, and broader comparisons to strengthen the contribution.

On balance, I concur with the reviewers that the paper’s technical novelty is limited, as it largely repurposes existing conformal prediction frameworks with only modest extensions. Nonetheless, it addresses a practically important and underexplored problem i.e. uncertainty quantification for irregularly sampled biomarker trajectories, and demonstrates that conformal prediction can be applied effectively in this clinically significant setting. In that sense, while the contribution may not be theoretically groundbreaking, it is a well-executed and meaningful application that offers real value.